# SpaceR: Reinforcing MLLMs in Video Spatial Reasoning

## Abstract

Video spatial reasoning, which involves inferring the underlying spatial structure from observed video frames, poses a significant challenge for existing Multimodal Large Language Models (MLLMs). This limitation stems primarily from 1) the absence of high-quality datasets for this task, and 2) the lack of effective training strategies to develop spatial reasoning capabilities. Motivated by the success of Reinforcement Learning with Verifiable Reward (RLVR) in unlocking LLM reasoning abilities, this work aims to improve MLLMs in video spatial reasoning through the RLVR paradigm. To this end, we introduce the **SpaceR** framework. First, we present **SpaceR-151k**, a dataset with 91k questions spanning diverse spatial reasoning scenarios with verifiable answers, and 60k samples for maintaining general multimodal understanding. Second, we propose **Spatially-Guided RLVR (SG-RLVR)**, a novel reinforcement learning approach that extends Group Relative Policy Optimization (GRPO) with a novel map imagination mechanism, which encourages the model to infer spatial layouts in the thinking process, thereby facilitating more effective spatial reasoning. Extensive experiments demonstrate that SpaceR achieves state-of-the-art performance on spatial reasoning benchmarks (e.g., VSI-Bench, STI-Bench, and SPAR-Bench), while showing competitive results on video understanding benchmarks (e.g., Video-MME, TempCompass, and LongVideoBench). Remarkably, SpaceR surpasses the advanced GPT-4o by 11.6% accuracy on VSI-Bench and is on par with the leading proprietary model Gemini-2.0-Flash, highlighting the effectiveness of our SpaceR-151k dataset and SG-RLVR in reinforcing spatial reasoning ability of MLLMs.

## 1 Introduction

Video spatial reasoning (Yang et al., 2024b) requires reconstructing 3D spatial layouts from sequences of observed frames. This task demands a higher-level cognitive ability than conventional video understanding tasks, such as video captioning (Venugopalan et al., 2015), video question answering (Antol et al., 2015), and temporal grounding (Gao et al., 2017), which typically require only recall of video content. Although recent advancements in Multimodal Large Language Models (MLLMs) have significantly improved performance on conventional video understanding (Chen et al., 2024d; Bai et al., 2025; Hurst et al., 2024), these models still struggle with video spatial reasoning (Li et al., 2025b; Yang et al., 2024b). This limitation stems mainly from two factors: 1) the absence of a high-quality dataset specifically designed for spatial reasoning, and 2) the reliance of most existing MLLMs on supervised fine-tuning (SFT) during post-training, which is insufficient for fostering deep reasoning capabilities.

In contrast to SFT, recent studies have demonstrated that Reinforcement Learning with Verifiable Rewards (RLVR) is more effective in enhancing the reasoning capabilities of both LLMs and MLLMs within pure-text (Guo et al., 2025) and multimodal tasks (Du et al., 2025; Feng et al., 2025). For example, DeepSeek-

R1-Zero (Guo et al., 2025) utilizes the Group Relative Policy Optimization (GRPO) algorithm to unlock the reasoning capabilities of LLMs. Video-R1 (Feng et al., 2025) introduces a large-scale multimodal understanding dataset Video-R1-260k and extends GRPO with a temporal reward for RLVR, which also achieves promising performance in video understanding benchmarks like MVBench (Li et al., 2023). Inspired by these findings, this work aims to advance MLLMs' spatial reasoning abilities in video through RLVR.

Specifically, we propose the **SpaceR** framework, which encompasses two key innovations: **First**, we introduce the SpaceR-151k dataset, which consists of 151k samples, including 91k spatial reasoning QA pairs (SR-91k) curated based on a 3D reconstruction dataset ScanNet (Dai et al., 2017), and 60k samples drawn from the general multimodal understanding dataset Video-R1-260k (Feng et al., 2025). In particular, SR-91k spans six spatial reasoning tasks (e.g., relative direction, object/room size, and appearance order), filling a critical gap in available resources. **Second**, we extend the GRPO (Shao et al., 2024) paradigm to enhance spatial reasoning by designing task-specific verifiable rewards for various QA formats (e.g., multiple choice, numerical). Furthermore, we introduce a novel map imagination mechanism. Unlike prior structured chain-of-thought methods (Feng et al., 2025; Li et al., 2025a) that simply produce step-by-step reasoning without explicit spatial grounding, this mechanism guides the model to explicitly generate a cognitive map, which is a structured representation of object positions in space, within specialized tags `<map>···</map>`. And a map reward is employed to accurately evaluate the quality of these inferred spatial layouts, which incentivizes models to think in space deeply.

Extensive experiments demonstrate that our SpaceR delivers consistent and significant performance gains across several challenging spatial reasoning benchmarks, including VSI-Bench (Yang et al., 2024b), STI-Bench (Li et al., 2025b), and SPAR-Bench (Zhang et al., 2025b), while presenting promising results in representative video understanding benchmarks like Video-MME (Fu et al., 2024), TempCompass (Liu et al., 2024b), and LongVideoBench (Wu et al., 2024). Notably, SpaceR achieves 45.6% accuracy on VSI-Bench (Yang et al., 2024b), surpassing the advanced proprietary model GPT-4o (Hurst et al., 2024) by 11.6% accuracy. These empirical results validate both the utility of the SpaceR-151k dataset and the effectiveness of SG-RLVR in unlocking spatial reasoning capabilities of MLLMs.

Our contributions are threefold: ♠ We introduce the SpaceR-151k dataset specifically designed for video spatial reasoning. It consists of questions spanning diverse spatial reasoning scenarios with verifable answers, addressing the scarcity of resources in this domain. ♥ We propose SG-RLVR, a spatially-guided reinforcement learning framework that integrates a novel map imagination mechanism. It encourages the model to explicitly incorporate spatial layouts into its reasoning process, thereby boosting video spatial reasoning. ♣ We conduct extensive evaluations across spatial reasoning and video understanding benchmarks, demonstrating that SpaceR achieves state-of-the-art spatial reasoning capabilities and promising generalizability in video understanding, validating the effectiveness of both the SpaceR-151k dataset and our SG-RLVR framework.

## 2 RELATED WORKS

### 2.1 VIDEO SPATIAL REASONING

Video understanding tasks like video captioning (Venugopalan et al., 2015), temporal grounding (Gao et al., 2017; Jin et al., 2022), and temporal perception (Lin et al., 2023), primarily focus on recalling or summarizing video content. For instance, video captioning necessitates models to generate relevant textual descriptions of the video based on human prompts. Recent advances in Multimodal Large Language Models (MLLMs) have significantly improved performance on these tasks. Unlike these conventional understanding tasks, video spatial reasoning requires models not only to perceive visual content but also to infer and reconstruct the spatial structure of entire scenes. It is worth noting that video spatial reasoning is crucial for the development of world models (Liu et al., 2024a; Cheng et al., 2024; Chen et al., 2024b) and embodied agents (An et al., 2022; Georgakis et al., 2022). Recent studies (Li et al., 2025b; Yang et al., 2024b; Zhang et al., 2025b) have

highlighted the persistent shortcomings of MLLMs on spatial reasoning, underscoring the need for further research in this area. A key limitation contributing to this gap is the scarcity of high-quality training data specifically tailored for video spatial reasoning, which we aim to address in this work.

## 2.2 REINFORCEMENT LEARNING WITH VERIFIABLE REWARD

Recent works like o1 (Jaech et al., 2024), DeepSeek-R1 (Guo et al., 2025), Kimi k1.5 (Team et al., 2025a) have demonstrated significant breakthroughs in enhancing the reasoning capabilities of large language models (LLMs) through Reinforcement Learning (RL). In particular, the Group Relative Policy Optimization (GRPO) (Shao et al., 2024) algorithm, applied in DeepSeek-R1, has revealed the strong potential of Reinforcement Learning with Verifiable Reward (RLVR) framework in equipping MLLMs with advanced reasoning capacity. Building on this foundation, several subsequent efforts (Liu et al., 2025b; Peng et al., 2025) have employed RLVR to boost visual reasoning performance. For example, Visual-RFT (Liu et al., 2025b) improved MLLMs in multimodal detection (Plummer et al., 2015), grounding (Yu et al., 2016), and classification (Russakovsky et al., 2015). LMM-R1 (Peng et al., 2025) empowers 3B MLLMs with strong reasoning abilities of mathematics through two-stage rule-based RL. Nevertheless, research specifically targeting video spatial reasoning remains underexplored. Motivated by this gap, our work seeks to design an effective reasoning paradigm to enhance MLLMs' capabilities in video spatial reasoning.

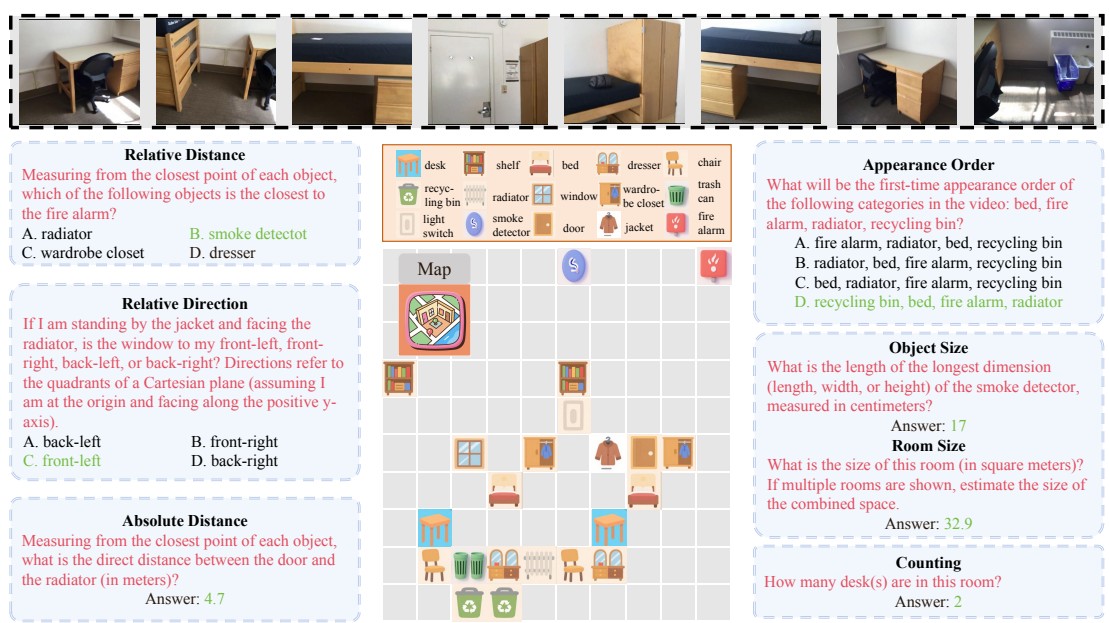

Figure 1: The overview of the Question-Answering examples from SR-91k, including multi-choice QA (e.g., relative distance, relative direction, appearance order) and numerical QA (e.g., object/room size, absolute distance, and counting), as well as the corresponding map for the video.

## 3 DATASET CONSTRUCTION

To address the scarcity of high-quality data for video spatial reasoning while maintaining general video understanding, we construct **SpaceR-151k**, a large-scale dataset consisting of two parts: 1) **SR-91k**, a tailored spatial reasoning dataset built upon the 3D scene reconstruction dataset ScanNet (Dai et al., 2017), and 2) 60k QA instances resampled from the general multimodal understanding dataset Video-R1-260k (Feng et al., 2025). The construction process follows three stages: data collection, QA generation, and data filtering. An overview of QA types and examples is provided in Figure 1.

**Data Collection.** 1) Spatial reasoning. We first parse ScanNet into a unified meta-information format, containing object categories, appearance indices, bounding box and other attributes, to facilitate QA generation. RGB frames are resampled at 24 FPS to form video clips. Besides, we construct a $10 \times 10$ map for each video to summarize the object distribution of the room, which is exemplified in Figure 1. Each object's coordinate is determined by the center point of its bounding box and projected onto a 2D map, which is detailed in Appendix A. 2) General understanding. To preserve general comprehension capabilities, we uniformly sample $60,000$ diverse QA instances from Video-R1-260k. This subset covers multiple QA types, including multi-choice, numerical, OCR, free-form, and regression.

**QA Generation.** Leveraging the parsed meta-information of ScanNet, we automatically generate the question-answering (QA) pairs for spatial reasoning tasks, which are categorized into multi-choice QA (e.g., relative distance, relative direction, and appearance order) and numerical QA (e.g., object/room size, absolute distance, and counting). The QA examples are presented in Figure 1, and the detailed generation process are as follows: 1) *Relative Distance*. For each video, we first identify unique objects, randomly select a target object and four candidate objects to be incorporated into the question template. Finally, the minimum Euclidean distance between each target and candidate is computed to determine the answer. 2) *Relative Direction*. Utilizing previous identified unique objects in the video, we randomly select three of them to be integrated in the question template. Relative directions are determined on the basis of their center points of bounding box. 3) *Appearance Order*. We record the first frame index where each object appears, and randomly sample four objects from them to generate the questions. The ordering is determined by their first frame indices. 4) *Object/Room Size*. Object size is defined as the longest dimension of an object computed from point clouds, and is converted to centimeters. Room size (in square meters) is estimated via the Alpha Shape algorithm[1]. 5) *Absolute Distance*. We uniformly sample points within the object bounding boxes and estimate the minimum Euclidean distance between two unique objects in the video. 6) *Counting*. We obtain the number of each object appearing in the video from the meta-information of ScanNet.

**Data Filtering.** To ensure the quality of the spatial reasoning QA pairs, we apply a series of filtering steps. First, we limit the number of QA pairs per video to promote scene diversity. For multi-choice QA, we randomly shuffle the positions of correct answers to balance answer distribution and eliminate position bias. In addition, we meticulously adjust the numerical value distribution in the numerical QA to prevent skewed or unrealistic value shifts. After filtering, we retain 91k high-quality QA pairs, forming the SR-91k dataset for spatial reasoning.

**Data Statistics.** The data statistics of SpaceR-151k are exhibited in Figure 2. This dataset comprises a total of $151,310$ samples, integrating 91k spatial reasoning QA pairs (SR-91k) with 60k instances drawn from the Video-R1-260k. SpaceR-151k features a diverse range of QA types: multi-choice, numerical, OCR, free-form, and regression, whose answers are verifiable. Above all, SpaceR-151k provides rich sources for both spatial reasoning and general understanding, which is the foundation of subsequent training. We visualize the duration distribution of videos from SR-91k in Figure 6.

---

[1] https://en.wikipedia.org/wiki/Alpha_shape.

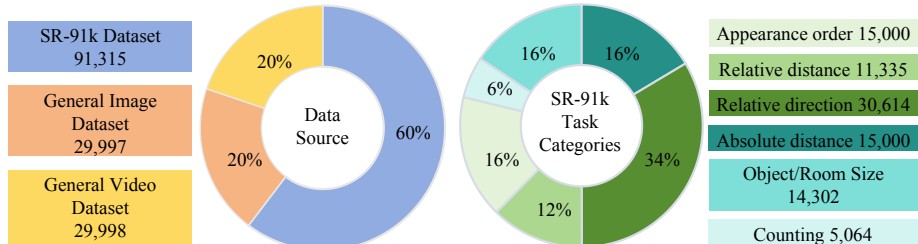

Figure 2: Data statistics of our SpaceR-151k. Left: the distribution of data sources. Right: the task category distribution within SR-91k.

## 4 SPATIALLY-GUIDED REINFORCEMENT LEARNING WITH VERIFIABLE REWARD

To reinforce video spatial reasoning in MLLMs, we propose a reinforcement learning framework named SG-RLVR, which builds on Group Relative Policy Optimization (GRPO) (Guo et al., 2025) by introducing verifiable reward functions tailored to diverse QA types and a novel map imagination mechanism to guide spatial reasoning.

### 4.1 VERIFIABLE REWARD FUNCTION

To supervise model outputs across multiple QA types, including multi-choice, numerical, OCR, free-form, and regression, we design a set of verifiable reward functions that assess either response format or correctness based on task-specific criteria.

**Format Reward.** To ensure the model responses adhere to a predefined structure, we define a format reward $R_{\text{format}}$ based on whether the model wraps its reasoning process and answer within <think>$\cdots$</think> and <answer>$\cdots$</answer> tags, respectively:

$$R_{format}(\hat{y}) = \begin{cases} 1, & \text{if } \hat{y} \text{ matches format,} \\ 0, & \text{otherwise.} \end{cases} \tag{1}$$

**Multi-choice Reward.** For multi-choice QA, the reward $R_{\text{mc}}$ is binary, based on exact match with the ground truth:

$$R_{mc}(\hat{y}, y) = \begin{cases} 1, & \text{if } \hat{y} = y, \\ 0, & \text{otherwise,} \end{cases} \tag{2}$$

where $\hat{y}$ is the model's response and $y$ is the ground truth.

**Numerical Reward.** To assess numerical values, we compute relative accuracy across varying confidence thresholds $\theta_i \in \{0.5, 0.55, \ldots, 0.95\}$. The numerical reward $R_{num}$ is defined as:

$$R_{num}(\hat{y}, y) = \frac{1}{N} \sum_{i=1}^{N} \mathbb{1}\left(\frac{|\hat{y} - y|}{y} \leq 1 - \theta_i\right), \tag{3}$$

$N$ is the number of confidence thresholds. Besides, for general multimodal understanding data from Video-R1-260k, we incorporate three additional reward functions: OCR, free-form, and regression rewards (Feng et al., 2025). The OCR reward is computed based on the Word Error Rate (WER)[2], which measures the

---

[2]https://en.wikipedia.org/wiki/Word_error_rate

edit distance between the predicted and reference text. The free-form reward is calculated as the average of ROUGE-1, ROUGE-2, and ROUGE-L scores (Lin, 2004) between the model's response and the ground truth. The regression reward is determined by the relative distance between the numerical values of response and ground truth.

## 4.2 MAP-BASED SPATIAL REASONING

Since previous RLVR frameworks like GRPO (Guo et al., 2025) lack explicit reward signals for spatial information comprehension when applied to video spatial reasoning, we propose a map imagination mechanism that encourages the model to think in space. Specifically, the model is guided to generate a $M \times M$ map to identify object distributions within the scene, supporting downstream reasoning and lead to more reliable answer. To evaluate the quality of the generated map, we design a novel map reward $R_{map}$ that provides precise quantitative feedback to facilitate spatial reasoning. Particularly, we first calculate the relative accuracy between predicted object and ground truth object by their relative distance $\frac{\sqrt{(x_{p,i}-x_{g,i})^2+(y_{p,i}-y_{g,i})^2}}{\sqrt{M^2+M^2}}$, where $M$ is the size of grid map, and average the relative accuracy across all objects to derive map reward $R_{map}$. Mathematically,

$$R_{map} = \sum_{i=1}^{k} \left( \frac{n_i}{\sum_{j=1}^{k} n_j} \times \left( 1 - \frac{\sqrt{(x_{p,i} - x_{g,i})^2 + (y_{p,i} - y_{g,i})^2}}{\sqrt{M^2 + M^2}} \right) \right), \tag{4}$$

where $k$ is the number of object categories, $n_i$ is the number of $i$-th object. $(x_{p,i}, y_{p,i})$ and $(x_{g,i}, y_{g,i})$ are the coordinates of the $i$-th object in the predicted map and ground truth map. To regulate the reasoning process, we introduce a length-based reward $R_l$ that encourages outputs to fall within a defined length range: $[l_{min}, l_{max}]$. This helps strike a balance between promoting sufficient reasoning and avoiding overthinking. $R_l$ is applied only when the model produces a correct answer within the desired length. Formally, the map imagination augmented reward $R_m$ is defined as:

$$R_m = \begin{cases} R_{format} + R_{task} + R_{map} + R_l, & \text{if } R_{task} = 1 \\ R_{format} + R_{task} + R_l, & \text{otherwise,} \end{cases} \tag{5}$$

where $task \in \{mc, num, ocr, free, reg\}$. This reward shaping ensures that when the model answers correctly and can properly understand the space of the indoor scene, it receives additional reward, pushing the optimization toward adopting a more spatial aware reasoning policy. The advantage $A_i$, representing the relative quality of the $i$-th response $o_i$, is computed over the updated rewards within each group of responses $[o_1, o_2, \cdots, o_G]$, where $G$ is the number of output responses. The final optimized policy $\pi_\theta$ is prevented from deviating far from the original model parameters $\pi_{\text{ref}}$ by adding a KL-divergence term $\mathcal{D}_{\text{KL}}(\cdot\|\cdot)$ to the following formulation:

$$A_i = \frac{R_m - \text{mean}(\{R_m\})}{\text{std}(\{R_m\})}. \tag{6}$$

The final policy update follows the clipped surrogate objective of GRPO:

$$J(\theta) = \mathbb{E}_{q,\{o_i\}} \left[ \frac{1}{G} \sum_{i=1}^{G} \min\left( \frac{\pi_\theta(o_i|q)}{\pi_{\theta_{\text{old}}}(o_i|q)} A_i, \text{clip}\left( \frac{\pi_\theta(o_i|q)}{\pi_{\theta_{\text{old}}}(o_i|q)}, 1-\epsilon, 1+\epsilon \right) A_i \right) - \beta \, \mathcal{D}_{\text{KL}}(\pi_\theta\|\pi_{\text{ref}}) \right] \tag{7}$$

where $\beta$ is a regularization coefficient, preventing excessive deviation from the reference policy during optimization, $\epsilon$ is a positive coefficient limits the policy updating degree.

# 5 EXPERIMENT

## 5.1 EXPERIMENTAL SETUPS

**Implementation Details.** 1) In the training stage, we adopt Qwen-2.5-VL-7B-Instruct as the base model and train for up to 2 epochs with a per-device batch size of 1. For each sample, 8 response candidates are generated, with a maximum completion length of $1,024$ tokens. $l_{min}$ and $l_{max}$ are set to 360 and 512, respectively. To balance efficiency and performance, videos are restricted to 16 frames, each processed at a resolution of $128 \times 28 \times 28$. The confidence thresholds number $N$ and the map size $M$ are both 10. 2) In the evaluation period, we prompt SpaceR to explicitly perform a step-by-step reasoning process on spatial reasoning benchmarks, while directly generating answers for video understanding benchmarks. The base model is only prompted for direct answers, as it is not trained for intermediate reasoning. The generation temperature is fixed at $0.01$. The maximum number of new tokens is set to $1,024$ when reasoning steps are included, and 128 for direct answer. The number of video frames is standardized to 32 during evaluation, and each frame is processed at a resolution of $448 \times 28 \times 28$.

**Benchmarks.** A diverse set of evaluation benchmarks is employed to comprehensively assess the model's capabilities in both spatial reasoning and video understanding. We conduct extensive evaluation on three spatial reasoning benchmarks (i.e., VSI-Bench (Yang et al., 2024b), STI-Bench (Li et al., 2025b), and SPAR-Bench (Zhang et al., 2025b)) and three video understanding benchmarks (i.e., Video-MME (Fu et al., 2024), TempCompass (Liu et al., 2024b), and LongVideoBench (Wu et al., 2024)). The detailed description and usage for each benchmark, as well as the evaluated baselines, are summarized in Appendix B.

| | #Params | Frames | VSI-Bench | STI-Bench | | SPAR-Bench | | | Video Understanding | | |
| | | | | Overall | SR_sub | Overall | Single-view | Multi-view | VM | TC | LV |
|---|---|---|---|---|---|---|---|---|---|---|---|
| *Closed-source Models* | | | | | | | | | | | |
| GPT-4o (Hurst et al., 2024) | - | - | 34.0 | 34.8 | 35.4 | 36.4 | 38.1 | 35.3 | 71.9 | 73.8 | 66.7 |
| Gemini 1.5 Pro | - | - | 48.8 | - | - | - | - | - | 75.0 | 67.1 | 64.0 |
| Gemini 2.0 Flash | - | - | 45.4 | 38.7 | 39.8 | - | - | - | - | - | - |
| Gemini 2.5 Pro | - | - | - | 40.9 | 40.5 | - | - | - | - | - | - |
| *Open-source Models* | | | | | | | | | | | |
| VideoLLaMA3-7B (Zhang et al., 2025a) | 7B | - | 29.7 | 26.9 | 27.2 | 32.6 | 33.4 | 32.1 | 66.2 | 68.1 | 59.8 |
| LLaVA-OneVision-7B (Li et al., 2024) | 7B | - | 32.4 | - | - | 31.2 | 33.1 | 29.9 | 58.2 | - | 56.3 |
| MiniCPM-V-2.6 (Yao et al., 2024) | 8B | - | 28.7 | 26.9 | 29.6 | 28.3 | 27.2 | 29.1 | 60.9 | 68.0 | 54.9 |
| Kimi-VL-A3B-Instruct (Team et al., 2025b) | 3B/16B | - | 37.4 | 34.2 | 31.4 | 34.9 | 33.4 | 36.0 | 62.3 | 70.3 | 58.0 |
| InternVL3-8B (Chen et al., 2024c) | 8B | - | 39.5 | 37.3 | 32.9 | 36.6 | 36.2 | 36.8 | 64.6 | 72.4 | 59.4 |
| Video-R1-7B-zero (Feng et al., 2025) | 7B | 32 | 31.8 | 32.9 | 29.8 | 28.6 | 30.6 | 27.3 | 54.9 | 71.0 | 53.7 |
| VideoChat-R1 (Li et al., 2025a) | 7B | 32 | 33.1 | 32.1 | 28.0 | 32.1 | 34.3 | 30.7 | 56.9 | 70.8 | 53.3 |
| SpaceQwen (Chen et al., 2024a) | 3B | 32 | 28.5 | 33.0 | 34.5 | 22.7 | 21.4 | 23.7 | 52.3 | 49.6 | 65.1 |
| Qwen2.5-VL-72B-Instruct (Bai et al., 2025) | 72B | 32 | 35.6 | 40.8 | 36.9 | 36.4 | 40.6 | 33.6 | 61.3 | 75.3 | 57.1 |
| Qwen2.5-VL-32B-Instruct (Bai et al., 2025) | 32B | 32 | 34.7 | 38.5 | 35.7 | 36.1 | 40.5 | 33.2 | 58.4 | 71.8 | 55.2 |
| Qwen2.5-VL-7B-Instruct (Bai et al., 2025) | 7B | 32 | 34.4 | 34.5 | 32.3 | 33.8 | 36.9 | 31.8 | 56.3 | 71.1 | 53.5 |
| SpaceR SFT | 7B | 32 | 41.6 | 30.2 | 27.3 | 33.3 | 35.1 | 32.0 | 57.6 | 69.3 | 54.3 |
| SpaceR SG-RLVR | 7B | 32 | 45.6 (↑11.2) | 37.0 (↑2.5) | 38.7 (↑6.4) | 37.6 (↑3.8) | 38.2 (↑1.3) | 37.1 (↑5.3) | 57.9 (↑1.6) | 71.4 (↑0.3) | 54.6 (↑1.1) |

Table 1: Evaluation results of base model Qwen2.5-VL-7B-Instruct, SpaceR, and other baselines on spatial reasoning benchmarks (VSI-Bench, STI-Bench, and SPAR-Bench), and video understanding benchmarks: **VM** (Video-MME), **TC** (TempCompass), and **LV** (LongVideoBench). SR_sub is a subset containing six spatial reasoning sub-tasks of STI-Bench.

## 5.2 MAIN RESULTS

The evaluation results on the six benchmarks are presented in Table 1. And we have the following observations and analyses.

**Overall Analysis.** Overall, our SpaceR consistently outperforms the base model Qwen2.5-VL-7B-Instruct across all benchmarks. In spatial reasoning benchmarks, SpaceR even surpasses the proprietary GPT-4o model, highlighting its superior spatial reasoning capabilities. Notably, SpaceR achieves a substantial accuracy

gain of 11.2 on VSI-Bench, a representative benchmark for spatial reasoning, highlighting its strengthened ability to model complex spatial relationships. Importantly, SpaceR attains nearly identical improvements on ScanNet (+11.6) and non-ScanNet (+11.0) subsets, indicating that its performance does not rely on domain memorization from ScanNet and is not inflated by the shared domain. Beyond spatial reasoning, SpaceR also generalizes well to video understanding tasks, achieving higher accuracy across all three benchmarks: Video-MME, TempCompass, and LongVideoBench, compared to Qwen2.5-VL-7B-Instruct. This indicates that the spatial reasoning enhancements and general multimodal understanding training samples contribute to broader video comprehension capabilities.

**SG-RLVR vs SFT.** We further compare the effectiveness of our SG-RLVR and SFT. While SFT yields localized improvements on benchmarks, such as VSI-Bench, Video-MME, and LongVideoBench, it leads to performance degradation on other benchmarks, indicating limited generalizability. In contrast, SG-RLVR consistently improves performance across spatial reasoning and video understanding benchmarks, highlighting its better generalizability. These results support the claim that "SG-RLVR generalizes, while SFT memorizes," establishing SG-RLVR as a more effective training paradigm for enhancing spatial reasoning in MLLMs. And we also provide a theoretical justification for this statement: SFT minimizes token-level likelihood, and the most direct way to reduce this loss is to reproduce target outputs. This objective inherently encourages the model to capture dataset-specific patterns, leading to memorization (Chu et al., 2025). In contrast, SG-RLVR optimizes format, task and map rewards, reinforcing any behavior that achieves the intended goals (i.e., producing a structured reasoning process that incorporates an optional accurate map to reach the correct answer). This reward structure encourages the model to learn task-level strategies that remain effective under distributional variations, thereby improving generalization.

**Impact of Data Sampling on Model Performance.** To improve training efficiency and model generalization, we apply a sample selection strategy to the SR-91k dataset by filtering out samples deemed too easy or too difficult. Using Qwen2.5-VL-7B-Instruct to generate 8 responses per sample, we categorize samples into all correct, partially correct, and all wrong, based on model response consistency, those all correct samples (low learning value) and all wrong samples (potential noise) are excluded. As illustrated in Figure 4(a), the remaining samples span a balanced range of task categories. Retraining SpaceR on this filtered dataset results in consistent performance gains across nearly all benchmarks, as shown in Figure 4(b). These findings suggest that targeted resampling prioritizes high-utility samples, thereby improving generalization in both spatial reasoning and video understanding. Additional analyses for the impact of thinking, data scale, and map quality, as well as the generalizability of our approach across model sizes and architectures, are provided in Appendix C.

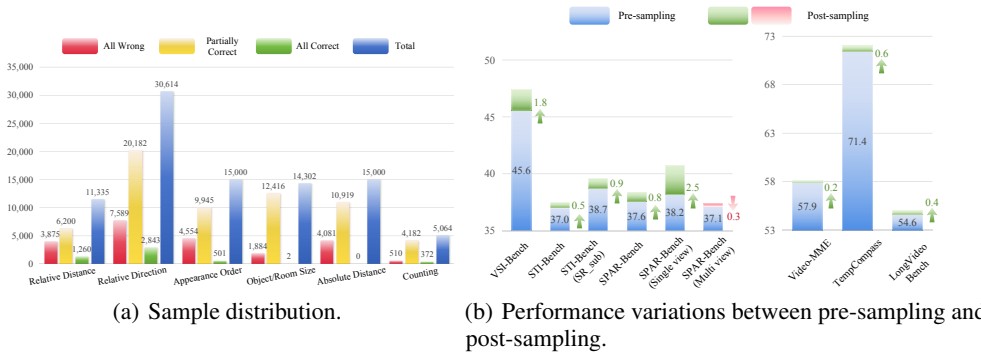

(a) Sample distribution.

(b) Performance variations between pre-sampling and post-sampling.

Figure 3: Sample distribution and Performance variations between pre-sampling and post-sampling.

| | Frames | Spatial Reasoning | | | | | | Video Understanding | | |
| | | VSI-Bench | STI-Bench | | SPAR-Bench | | | VM | TC | LV |
| | | | Overall | SR_sub | Overall | Single-view | Multi-view | | | |
|---|---|---|---|---|---|---|---|---|---|---|
| w/o-map imagination | 32 | 43.9 | 34.0 | 33.3 | 34.1 | 34.7 | 33.8 | 56.9 | **71.9** | 52.8 |
| w/o-general data | 32 | **46.9** | **37.1** | 37.2 | 37.1 | 35.5 | **39.5** | 56.3 | 71.0 | 53.6 |
| w/o-SR data | 32 | 26.2 | 34.2 | 33.9 | 30.2 | 30.7 | 29.5 | 57.4 | 70.5 | 52.9 |
| w/o-length reward | 32 | 44.0 | 35.8 | 36.5 | 35.9 | 36.6 | 35.5 | 56.7 | 71.2 | 54.0 |
| SpaceR SG-RLVR | 32 | 45.6 | 37.0 | **38.7** | **37.6** | **38.2** | 37.1 | **57.9** | 71.4 | **54.6** |

Table 2: Ablation results of SpaceR SG-RLVR, where the best results are in boldface.

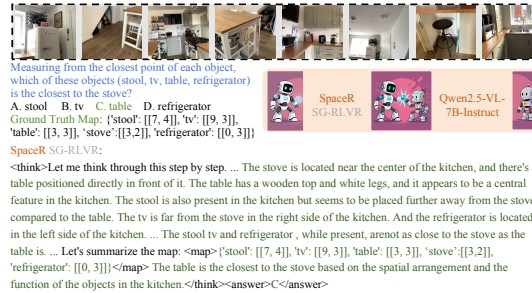

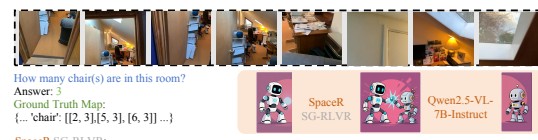

(a) Comparison on object relative distance task.

(b) Comparison on object counting task.

Figure 4: Two samples from VSI-Bench (Yang et al., 2024b), as well as the corresponding responses from SpaceR SG-RLVR and Qwen2.5-VL-7B-Instruct (Bai et al., 2025).

## 5.3 ABLATION STUDY

To explore the contributions of individual components in our method, we introduce four variants of our SpaceR: 1) w/o-map imagination, which eliminates the map imagination mechanism in the training and inference stage. 2) w/o-general data, which excludes 60k general multimodal understanding data in the training stage. 3) w/o-SR data, which removes SR-91k data in the training process. 4) w/o-length reward, which omits the length reward during training. The ablation results are presented in Table 2, based on which we have the following findings. a) SpaceR consistently outperforms w/o-map imagination on three spatial reasoning benchmarks, which validates the advantage of map imagination mechanism to guide MLLMs in understanding spatial information. b) SpaceR exceeds w/o-general data on video understanding benchmarks, which proves the effectiveness of the 60, 000 general multimodal understanding training data. c) w/o-SR data shows significantly lower performance on spatial reasoning benchmarks compared to SpaceR, emphasizing the critical importance of SR-91k dataset in enhancing spatial reasoning capabilities. d) w/o-length reward underperforms SpaceR, underscoring the utility of length reward in fostering efficient reasoning.

## 5.4 QUALITATIVE ANALYSIS

To get an intuitive understanding on the advancement of SpaceR in video spatial reasoning, we present two cases from VSI-Bench in Figure 4. In case (a), SpaceR demonstrates a clear qualitative superiority over Qwen2.5-VL-7B through structured reasoning, explicit spatial mapping, and an accurate conclusion. SpaceR correctly identifies the "table" as the object closest to the "stove", supporting its decision with a cognitive map that simulates the spatial layouts of the scene. In contrast, Qwen2.5-VL-7B relies on shallow, assumption-based heuristics and incorrectly selects the "stool", reflecting a lack of verifiable spatial reasoning.

This emphasizes SpaceR's enhanced spatial awareness and reasoning depth, enabled by its map imagination mechanism. Similarly, in case (b), SpaceR also beats Qwen2.5-VL-7B on a "chair-counting" task. By reasoning across multiple frames and accounting for partially occluded objects, SpaceR accurately concludes the presence of at least three chairs and reinforces its answer with an accurate cognitive map. In contrast, Qwen2.5-VL-7B underestimates the count, providing a wrong answer of two chairs. Together, these cases prove the SpaceR's improved ability of spatial reasoning. The failure cases are provided in Appendix C.7.

## 6 CONCLUSION

In this work, we introduce SpaceR, a novel framework designed to enhance video spatial reasoning capabilities. To this end, we construct SpaceR-151k, a comprehensive dataset that includes 91k high-quality spatial reasoning QA paris (SR-91k) and 60k samples for general video understanding. Building upon this dataset, we propose Spatially-Guided Reinforcement Learning with Verifiable Reward (SG-RLVR), a novel reinforcement framework, which integrates task-specific reward functions and a map imagination mechanism to guide models in spatial layout inference and foster structured spatial reasoning. Extensive evaluations across three spatial reasoning benchmarks and video understanding benchmarks validate the effectiveness and generalizability of SpaceR. We hope that SpaceR serves as a solid foundation for advancing research in video spatial reasoning and inspires further exploration into reasoning-aware training for MLLMs.

## REPRODUCIBILITY STATEMENT

To ensure reproducibility, we provide dataset curation details in Section 3 and Appendix A. The training and inference settings are described in Section 5.1, while hardware specifications and documentation/licensing information are presented in Appendix B.3 and Appendix D.

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

## A    MORE DETAILS FOR DATA CONSTRUCTION

| | |
|---|---|
| Relative Distance | Measuring from the closest point of each object, which of these objects ({object a}, {object b}, {object c}, {object d}) is the closest to the {target object}? |
| Relative Direction | 1. If I am standing by the {positioning object} and facing the {orienting object}, is the {querying object} to the left or the right of the {orienting object}?
2. If I am standing by the {positioning object} and facing the {orienting object}, is the {querying object} to my left, right, or back? An object is to my back if I would have to turn at least 135 degrees in order to face it.
3. If I am standing by the {positioning object} and facing the {orienting object}, is the {querying object} to my front-left, front-right, back-left, or back-right? Directions refer to the quadrants of a Cartesian plane (assuming I am at the origin and facing the positive y-axis). |
| Appearance Order | What will be the first-time appearance order of the following categories in the video: {choice a}, {choice b}, {choice c}, {choice d}? |
| Object/Room Size | 1. What is the length of the longest dimension (length, width, or height) of the {object}, measured in centimeters?
2. What is the size of this room (in square meters)? If multiple rooms are shown, estimate the size of the combined space. |
| Absolute Distance | Measuring from the closest point of each object, what is the direct distance between the {object 1} and the {object 2} (in meters)? |
| Counting | How many {object}(s) are in this room? |

Figure 5: Question templates for QA pairs of SR-91k.

**Map Construction.** Our map construction process begins by rasterizing the scene into a $10 \times 10$ grid. We then identify the center point of each object using the 3D point-cloud annotations provided in ScanNet. Each object is subsequently projected onto the 2D map based on the coordinates of its center point. This results in a map where each object is assigned a specific (x, y) coordinate, enabling efficient spatial representation and reasoning. In addition, to define the map's boundary, we first identify the floor level (where the z-axis is 0) within the 3D point-cloud data. We approximate this floor region as a rectangle to determine the map's boundary. For each object, we then sample the corresponding 3D point-cloud and compute its 3D bounding box to accurately identify the object's boundary within the scene. Finally, the map is represented in a dictionary format for structured spatial reference.

**QA Generation.** To format the generated QA pairs, we incorporate the corresponding objects into the specified question templates, which are presented in Figure 5.

**Data Filtering.** We remove the QA pairs that involve some noisy objects (e.g., "wall", "floor", and "ceiling"). And we also drop the numerical QA pairs, where the objects are too small to identify. Considering VSI-Bench (Yang et al., 2024b) is partially built on ScanNet (Dai et al., 2017), we exclude the overlapped videos in our SR-91k to ensure fair evaluation.

## B    MORE DETAILS OF EXPERIMENTAL SETUPS

### B.1    BENCHMARKS DESCRIPTION

- **VSI-Bench** (Yang et al., 2024b) is a comprehensive benchmark for evaluating the visual-spatial intelligence of Multimodal Large Language Models (MLLMs). It comprises over $5,000$ question-

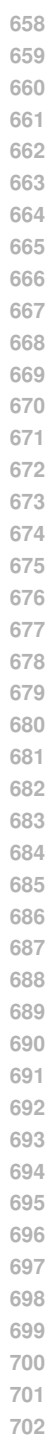

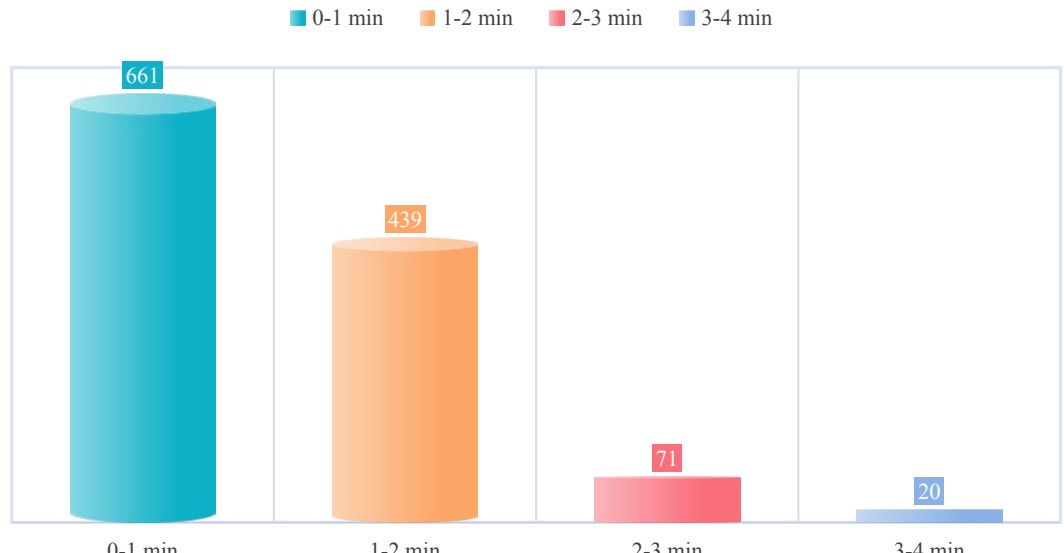

Figure 6: Video duration distribution of SR-91k.

answer pairs across 288 real-world indoor scene videos, covering diverse environments such as homes, offices, and factories, and is specifically designed to assess spatial reasoning capabilities.

- **STI-Bench** (Li et al., 2025b) evaluates the spatial understanding abilities of MLLMs using real-world videos spanning desktop, indoor, and outdoor scenarios. It includes eight challenging tasks, with the subset SR_sub, which contains more than $2,000$ QA pairs across six sub-tasks (i.e., Dimensional Measurement, Displacement & Path Length,Ego-Centric Orientation, Spatial Relation, Speed & Acceleration, Trajectory Description), being most relevant to our focus on spatial reasoning.

- **SPAR-Bench** (Zhang et al., 2025b) is specifically designed to measure the spatial understanding of MLLMs. It contains over $7,000$ QA pairs covering a spectrum of tasks from basic perception to complex spatial reasoning. The benchmark is further divided into single-view and multi-view settings, allowing for comprehensive assessment across varying spatial contexts.

- **Video-MME** (Fu et al., 2024) serves as a comprehensive benchmark for evaluating general video understanding in MLLMs. It includes 900 videos and $2,700$ high-quality multi-choice questions (three per video), spanning a wide range of scenarios and tasks. We exclude the subtitles of videos in the evaluation.

- **TempCompass** (Liu et al., 2024b) focuses on temporal perception in MLLMs. It consists of $410$ videos and $7,540$ questions designed to evaluate models' understanding of temporal dynamics.

- **LongVideoBench** (Wu et al., 2024) is a benchmark for long-context multimodal video understanding. It features $6,678$ carefully constructed multi-choice questions derived from videos of varying durations, extending up to one hour, and encompasses diverse real-world themes. We utilize the validations set of it and remove the subtitles of videos.

## B.2 BASELINES DESCRIPTION

- **GPT-4o** (Hurst et al., 2024) is a state-of-the-art MLLM developed by OpenAI, exhibiting strong performance across a variety of vision-language tasks.

- **Gemini 1.5 Pro, Gemini 2.0 Flash, Gemini 2.5 Pro** are advanced MLLMs from Google's Gemini family[3]. These models have shown leading performance across several video understanding benchmarks (e.g., Video-MME (Fu et al., 2024), and LongVideoBench (Wu et al., 2024)). Gemini 2.0 Flash and Gemini 2.5 Pro, in particular, exhibit enhanced abilities in complex reasoning tasks.

- **VideoLLaMA3-7B** (Zhang et al., 2025a) is an MLLM tailored for both image and video understanding. It adopts Qwen2.5-7B (Yang et al., 2024a) as its language backbone and integrates siglip-so400m-patch14-384 (Zhai et al., 2023) as the vision encoder.

- **LLaVA-OneVision-7B** (Li et al., 2024) represents a strong advancement in open-source multimodal language models (LMMs), combining the Qwen2 (Yang et al., 2024a) language backbone with the SigLIP (Zhai et al., 2023) vision encoder. This integration pushes the performance boundaries of open LMMs, particularly in tasks requiring fine-grained visual understanding.

- **MiniCPM-V-2.6** (Yao et al., 2024) is developed based on SigLIP-400M (Zhai et al., 2023) and Qwen2-7B (Yang et al., 2024a), and introduces enhanced capabilities for multi-image and video understanding. Its architectural improvements and task-specific design make it a competitive model for complex multimodal understanding tasks.

- **Kimi-VL-A3B-Instruct** (Team et al., 2025b) is an efficient open-source MLLM based on a Mixture-of-Experts (MoE) architecture. It incorporates the Moonlight (Liu et al., 2025a) MoE language model and the high-resolution MoonViT (Team et al., 2025b) vision encoder.

- **InternVL3-8B** (Chen et al., 2024c) is a high-performing open-source MLLM that combines InternViT-300M-448px-V2_5 (Chen et al., 2024d) as the vision encoder with Qwen2.5-7B (Yang et al., 2024a) as the LLM backbone.

- **Video-R1-7B-zero** (Feng et al., 2025) is an RLVR-based MLLM built upon Qwen2.5-VL-7B-Instruct, leveraging T-GRPO to enhance its video reasoning capability.

- **VideoChat-R1** (Li et al., 2025a) is a video MLLM built on Qwen2.5-VL-7B-Instruct, employing RLVR to achieve state-of-the-art spatiotemporal perception.

- **SpaceQwen** (Chen et al., 2024a) equips Qwen2.5-VL-3B with quantitative spatial reasoning capabilities using spatial reasoning data based on real-world images.

- **Qwen2.5-VL-7B-Insturct, Qwen2.5-VL-32B-Insturct, Qwen2.5-VL-72B-Insturct** (Bai et al., 2025) are part of the Qwen2.5-VL series, which combine the Qwen2.5 (Yang et al., 2024a) language model with a redesigned Vision Transformer (ViT) architecture for enhanced visual grounding and understanding.

## B.3 HARDWARE USAGE

Model training is conducted under 8 L20 80 GiB GPUs or 4 A800 80 GiB GPUs. Model is evaluated under 4 L20 80 GiB GPUs.

---

[3] https://aistudio.google.com.

## C   MORE EMPIRICAL RESULTS AND ANALYSES

In this section, we supply more comparison results, analyses for impact of thinking, data scale, and map quality on model performance, as well as the generalizability of our SG-RLVR across model sizes and architectures.

| | #Params | Frames | VSI-Bench | STI-Bench | | SPAR-Bench | | | Avg. Tokens | VM | TC | LV | Avg. Tokens |
|---|---|---|---|---|---|---|---|---|---|---|---|---|---|
| | | | | Overall | SR_sub | Overall | Single-view | Multi-view | | | | | |
| Kimi-VL-Thinking (Team et al., 2025b) | 3B | 16 | 32.5 | 31.6 | 29.8 | 27.3 | 26.8 | 27.6 | - | - | - | - | - |
| SpaceR-Kimi | 3B | 16 | 43.5 | 35.2 | 32.7 | 32.1 | 32.2 | 32.0 | - | - | - | - | - |
| Qwen2.5-VL-3B-Instruct (Bai et al., 2025) | 3B | 32 | - | - | - | - | - | - | - | - | - | - | - |
| + non-think | - | - | 26.7 | 36.7 | 37.5 | 25.4 | 25.3 | 25.5 | - | 52.4 | 65.7 | 49.7 | - |
| + think | - | - | 25.9 (↓0.8) | 34.3 (↓2.4) | 37.2 (↓0.3) | 26.8 (↑1.4) | 26.6 (↑1.3) | 26.9 (↑1.4) | 66.9 | 52.0 (↓0.4) | 63.2 (↓2.4) | 49.0 (↓0.7) | 0.3 |
| SpaceR-Tiny SFT | 3B | 32 | - | - | - | - | - | - | - | - | - | - | - |
| + non-think | - | - | 34.8 | 33.0 | 36.5 | 24.8 | 24.5 | 24.9 | - | 53.4 | 63.8 | 50.7 | - |
| SpaceR-Tiny SG-RLVR | 3B | 32 | - | - | - | - | - | - | - | - | - | - | - |
| + non-think | - | - | 40.5 | 36.6 | 38.7 | 30.1 | 30.7 | 29.6 | - | 52.9 | 66.4 | 50.1 | - |
| + think | - | - | 41.2 (↑0.7) | 37.8 (↑1.2) | 40.1 (↑1.4) | 30.9 (↑0.8) | 31.4 (↑0.7) | 30.6 (↑1.0) | 274.2 | 51.6 (↓1.3) | 65.4 (↓1.0) | 49.4 (↓0.7) | 237.1 |
| Video-R1-SFT | 7B | 32 | - | - | - | - | - | - | - | - | - | - | - |
| + think | - | - | - | - | - | - | - | - | - | 55.4 | 69.9 | 52.3 | - |
| SpaceR (Video-R1-SFT) SG-RLVR | 7B | 32 | - | - | - | - | - | - | - | - | - | - | - |
| + non-think | - | - | - | - | - | - | - | - | - | 57.1 | 71.7 | 53.0 | - |
| + think | - | - | - | - | - | - | - | - | - | 60.6 (↑3.5) | 73.5 (↑1.8) | 54.6 (↑1.6) | - |
| Qwen2.5-VL-7B-Instruct (Bai et al., 2025) | 7B | 32 | - | - | - | - | - | - | - | - | - | - | - |
| + non-think | - | - | 34.4 | 34.5 | 32.3 | 33.8 | 36.9 | 31.8 | - | 56.3 | 71.1 | 53.5 | - |
| + think | - | - | 30.2 (↓4.2) | 33.2 (↓1.3) | 34.4 (↑2.1) | 31.6 (↓2.2) | 31.2 (↓5.7) | 31.8 (-0.0) | 104.0 | 54.0 (↓2.3) | 68.1 (↓3.0) | 46.6 (↓6.9) | 68.6 |
| SpaceR SG-RLVR | 7B | 32 | - | - | - | - | - | - | - | - | - | - | - |
| + non-think | - | - | 45.0 | 36.7 | 34.8 | 36.1 | 36.2 | 36.0 | - | 57.9 | 71.4 | 54.6 | - |
| + think | - | - | 45.6 (↑0.6) | 37.0 (↑0.3) | 38.7 (↑3.9) | 37.6 (↑1.5) | 38.2 (↑2.0) | 37.1 (↑1.1) | 345.6 | 56.4 (↓1.5) | 70.0 (↓1.4) | 51.7 (↓2.9) | 265.7 |

Table 3: Comparison of SpaceR-Kimi with its base model Kimi-VL-Thinking, as well as SpaceR-Tiny, Qwen2.5-VL-3B-Instruct, SpaceR , and Qwen2.5-VL-7B-Instruct on spatial reasoning and video understanding benchmarks. non-think means direct answer generation, while think refers to engaging thinking during inference. **Avg. Tokens** indicates the average token count of thinking process.

### C.1   IMPACT OF THINKING ON MODEL PERFORMANCE

To assess the effect of explicitly engaging the thinking process during inference, we compare model performance under two modes: *non-think*, which outputs answers directly, and *think*, which includes a structured reasoning process. As shown in Table 3, models not explicitly trained to reason, such as Qwen2.5-VL-3B-Instruct, Qwen2.5-VL-7B-Instruct, and SpaceR-Tiny SFT, exhibit a significant performance drop across most benchmarks in *think* mode. This degradation is expected, since these models lack sufficient reasoning capability and often generate shorter, less informative reasoning traces, which could be reflected by their lower average token counts. In contrast, our SpaceR SG-RLVR demonstrates consistent gains on spatial reasoning benchmarks when utilizing the *think* mode. This indicates that our SG-RLVR method helps the model develop useful reasoning strategies. And we note that performance degradation in general video understanding under think mode is common. Compared to Qwen2.5-VL-7B-Instruct, SpaceR shows substantially smaller performance degradation when activating think mode, indicating that SpaceR mitigates, rather than amplifies, the negative impact of thinking in general video understanding. When initialized with Video-R1-SFT (Feng et al., 2025), which learns effective reasoning traces for general video understanding during SFT, SpaceR exhibits promising performance gains of general video understanding in think mode. This validates the practicality of our framework in enhancing general video understanding beyond video spatial reasoning.

### C.2   HOW THINKING ENHANCES SPATIAL REASONING.

The specific outputs from the thinking process can enhance reasoning in two key ways. On the one hand, spatial words (e.g., "near", "behind", "far", and other spatial related terms) play an important role in guiding the model toward accurate spatial reasoning, which improves overall performance on spatial reasoning benchmarks. In Table 4, we present the spatial word frequency (per sample) and the corresponding accuracy of spatial reasoning benchmarks. The results indicate a positive correlation between the frequency of spatial

| Model | VSI-Bench | | STI-Bench | | SPAR-Bench | |
|---|---|---|---|---|---|---|
| | Freq ↑ | Acc (%) | Freq ↑ | Acc (%) | Freq ↑ | Acc (%) |
| Qwen2.5-VL-7B-Instruct | 0.0 | 34.4 | 0.0 | 34.5 | 0.0 | 33.8 |
| SpaceR-1000 iters | 3.1 | 40.5 | 1.7 | 35.7 | 2.2 | 35.4 |
| SpaceR-3000 iters | 6.3 | 43.2 | 5.8 | 36.5 | 3.9 | 37.1 |
| SpaceR | 8.7 | 45.6 | 7.2 | 37.0 | 4.3 | 37.6 |

Table 4: Average spatial word frequency (Freq) in responses and accuracy (%) across the three spatial reasoning benchmarks.

words in the model's reasoning process and the performance on spatial reasoning tasks. On the other hand, we observe specific self-reflection patterns within the model's thinking process, e.g., "Now, let's verify this by checking the spatial arrangement". These reflective moments are beneficial for producing more accurate and coherent answers, as they reflect the model's ability to perform internal validation and refine its reasoning.

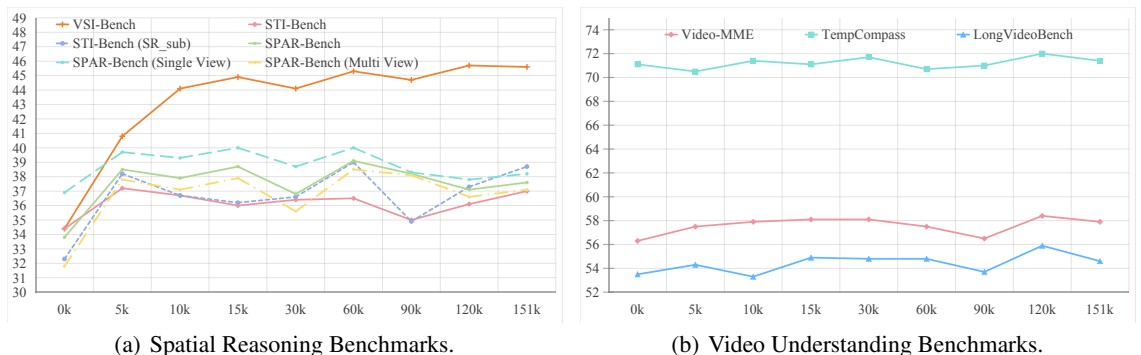

(a) Spatial Reasoning Benchmarks.  (b) Video Understanding Benchmarks.

Figure 7: Performance variations with progressively increasing data scale on spatial reasoning and video understanding benchmarks.

## C.3 IMPACT OF DATA SCALE ON MODEL PERFORMANCE.

To examine the relationship between data scale and model performance, we train SpaceR on six progressively larger subsets of the SpaceR-151k dataset. The results, presented in Figure 7, reveal two key observations. First, SpaceR demonstrates notable performance improvements on spatial reasoning benchmarks such as VSI-Bench and SPAR-Bench (Single View) even with small-scale subsets (e.g., 5k–15k), highlighting the strong data efficiency of our SG-RLVR training paradigm. Second, while performance generally plateaus or slightly fluctuates as the data scale increases to 30k, a substantial jump is observed when training on the full dataset (151k), especially for VSI-Bench, where accuracy surpasses 45%. These findings suggest that our method not only benefits from larger training sets but also exhibits strong generalization from limited supervision, underscoring its effectiveness in both low-resource and full-scale settings.

| Map type | Map accuracy | Appearance order | Absolute distance | Counting | Relative distance | Object size | Room size | Route planning | Relative direction | Overall |
|---|---|---|---|---|---|---|---|---|---|---|
| Randomly generated | 10.5 | 44.0 | 28.9 | 51.5 | 37.9 | 56.1 | 33.2 | 28.4 | 44.2 | 40.5 |
| Model-generated | 42.1 | 50.3 | 31.8 | 62.2 | 40.0 | 59.4 | 46.0 | 32.0 | 43.9 | 45.6 |
| Ground truth | 100.0 | 56.5 | 33.6 | 64.1 | 41.8 | 62.3 | 46.7 | 32.5 | 44.7 | 47.7 |

Table 5: Performance comparison of SpaceR SG-RLVR on VSI-Bench across different map types.

## C.4 IMPACT OF MAP QUALITY ON MODEL PERFORMANCE.

To analyze the impact of map imagination quality on spatial reasoning tasks, we conduct a comparison experiment on VSI-Bench, using three types of maps during inference: ground truth maps, model-generated maps, and randomly generated maps. As shown in Table 5, we observe a positive correlation between map accuracy and task performance, indicating that higher-fidelity spatial representations directly enhance spatial reasoning performance. These findings further demonstrate the importance of SpaceR's map imagination mechanism in enhancing spatial reasoning. To explore whether cognitive maps align with human spatial intuition, we randomly selected 50 correctly answered test samples, and found that 41 of them could be solved by three human annotators directly using the cognitive map, with substantial agreement (above 70%) (Gwet, 2014). This confirms that the generated maps are largely consistent with human spatial intuition.

| Map size | Map accuracy | Appearance order | Absolute distance | Counting | Relative distance | Object size | Room size | Route planning | Relative direction | Overall |
|---|---|---|---|---|---|---|---|---|---|---|
| $5 \times 5$ | 53.7 | 40.0 | 27.1 | 52.5 | 37.9 | 55.0 | 36.9 | 30.4 | 46.4 | 40.8 |
| $10 \times 10$ | 42.1 | 50.3 | 31.8 | 62.2 | 40.0 | 59.4 | 46.0 | 32.0 | 43.9 | 45.6 |
| $15 \times 15$ | 30.6 | 49.0 | 28.4 | 61.3 | 40.6 | 54.7 | 40.4 | 28.9 | 46.7 | 43.7 |
| $20 \times 20$ | 24.5 | 34.1 | 30.9 | 61.8 | 37.6 | 57.4 | 38.9 | 28.4 | 44.9 | 41.1 |

Table 6: Performance comparison of SpaceR SG-RLVR on VSI-Bench across different map sizes.

## C.5 IMPACT OF MAP SIZE ON MODEL PERFORMANCE.

To explore the effect of map size on spatial reasoning, we compare model performance across different map sizes in Table 6. Compared to $10 \times 10$ map, we observe that increasing map size substantially raises the difficulty of map imagination, resulting in lower map accuracy and degraded performance. Conversely, a coarser ($5 \times 5$) map is easier to imagine but introduces excessive spatial ambiguity and occlusions, limiting its representational capacity. Therefore, we adopt a $10 \times 10$ map as a balanced choice between spatial granularity and imagination difficulty.

## C.6 GENERALIZATION ACROSS MODEL SIZES AND ARCHITECTURES.

To examine the scalability of our SG-RLVR framework across different model sizes, we fine-tune both Qwen2.5-VL-3B-Instruct and Qwen2.5-VL-7B-Instruct (Bai et al., 2025) on the SpaceR-151k dataset using both supervised fine-tuning (SFT) and our proposed SG-RLVR method. As shown in Table 3, the base models exhibit limited spatial reasoning capabilities, with noticeable performance drops in the *think* mode, where they, especially Qwen2.5-VL-3B-Instruct, struggle to produce coherent reasoning and often revert to direct answer outputs. Although SFT leads to modest gains on benchmarks such as VSI-Bench, it fails to enhance generalization and does not substantially improve reasoning ability. In contrast, our SpaceR-Tiny SG-RLVR and SpaceR SG-RLVR consistently outperform the base models and their SFT counterparts across multiple spatial reasoning benchmarks, particularly in the *think* mode. Meanwhile, they show promising generalizability on video understanding benchmarks. These results confirm both the effectiveness and scalability of the SG-RLVR framework in enhancing spatial reasoning across model sizes.

To explore the generalizability of SG-RLVR across model architectures, we adopt Kimi-VL-Thinking (Team et al., 2025b) as the base model, a Mixture-of-Experts (MoE) vision-language model that activates only 2.8B parameters in its language decoder, in contrast to the dense VLM Qwen2.5-VL. As shown in Table 3, SpaceR-

Kimi achieves notable performance gains on three challenging spatial reasoning benchmarks, demonstrating SG-RLVR's effectiveness across heterogeneous architectures for enhancing video spatial reasoning.

(a) Failure Case1.

(b) Failure Case2.

Figure 8: Two failure cases of SpaceR SG-RLVR on VSI-Bench (Yang et al., 2024b).

## C.7 FAILURE CASES OF SPACER.

As illustrated in Figure 8, we identify two representative failure modes of SpaceR in spatial reasoning. 1) Visual Perception Error. For the query "How many sofa(s) are in this room?", the model predicted 2 instead of the ground truth 4. Although the cognitive map included two sofa instances, it failed to recognize the others, revealing limitations in object perception and enumeration. This typically stem from challenging visual conditions such as occlusion, low resolution, cluttered layouts, or small object instances. Under these conditions, the vision encoder may fail to detect certain objects, and these missed detections lead to missed objects of the cognitive map, as spatial understanding relies predominantly on the encoder's visual grounding (Zhang et al., 2025c). 2) Location Identification Error. For the query "Which of these objects is closest to the dishwasher?", the model incorrectly answered "stool". This error stems from mislocalized object positions in the cognitive map, which led to inaccurate distance estimation. This primarily arises from imprecise spatial grounding when the model infers object positions from limited visual cues, particularly in scenes with ambiguous perspectives or complex layouts. And the actionable solutions are: a) Incorporating challenging visual perception data to strengthen the vision encoder of Qwen2.5-VL, establishing a more robust perceptual foundation before learning spatial reasoning. and b) Integrating multi-view or depth-aware signals to improve the accuracy of spatial localization.

## D DOCUMENTATION AND LICENSING

The SpaceR-151k dataset includes annotations in JSONL format, along with associated images and videos. The SpaceR model adopts the same architecture as Qwen2.5-VL-7B-Instruct. Both the dataset and the model are released under the CC BY-NC 4.0 license[4] and are intended for academic research purposes only. Additionally, the ScanNet (Dai et al., 2017) dataset has been released under MIT license.

---

[4] https://creativecommons.org/licenses/by-nc/4.0/

# E   LLM USAGE

We employ large language models (e.g., ChatGPT[5]) to assist with text polishing and grammar checking.

---

[5]https://chatgpt.com

