# OpenReview forum: "SpaceR: Reinforcing MLLMs in Video Spatial Reasoning"
_ICLR.cc/2026/Conference — Submitted to ICLR 2026_

### Official Review · Reviewer_sGvR · 2025-10-15

**Soundness:** 3
**Presentation:** 2
**Contribution:** 3
**Rating:** 6
**Confidence:** 4

**Summary:**

This paper addresses the limitations of existing Multimodal Large Language Models (MLLMs) in video spatial reasoning (inferring 3D spatial structures from video frames), which arise from the lack of high-quality task-specific datasets and effective training strategies. Inspired by Reinforcement Learning with Verifiable Reward (RLVR), the authors propose the SpaceR framework, consisting of two core innovations: SpaceR-151k Dataset: A large-scale dataset with 91k spatial reasoning QA pairs (SR-91k, derived from the 3D ScanNet dataset, covering 6 tasks like relative direction and object size) and 60k general multimodal samples (from Video-R1-260k) to preserve general video understanding.
Spatially-Guided RLVR (SG-RLVR): An RL approach extending Group Relative Policy Optimization (GRPO) with a map imagination mechanism and task-specific verifiable rewards (e.g., format, multi-choice, numerical, map rewards).
Extensive experiments show SpaceR achieves state-of-the-art performance on spatial reasoning benchmarks (VSI-Bench, STI-Bench, SPAR-Bench)—surpassing GPT-4o by 11.6% on VSI-Bench—and competitive results on general video understanding benchmarks (Video-MME, TempCompass, LongVideoBench). It also matches the proprietary Gemini-2.0-Flash, validating the framework’s effectiveness.

**Strengths:**

Targeted Dataset Solution: SpaceR-151k fills a critical gap in video spatial reasoning resources by providing 91k high-quality, verifiable QA pairs across diverse spatial tasks (e.g., relative distance, room size), addressing the field’s data scarcity.

Innovative RL Enhancement: The map imagination mechanism explicitly guides models to infer spatial layouts, a novel design that strengthens deep spatial reasoning (ablation studies confirm it boosts performance on spatial benchmarks).

Strong Generalization: SG-RLVR outperforms supervised fine-tuning (SFT) across both spatial reasoning and general video tasks, avoiding SFT’s "memorization" limitation and demonstrating broader applicability.

Comprehensive Validation: Experiments cover 3 spatial and 3 general video benchmarks, with comparisons to state-of-the-art models (GPT-4o, Gemini series, open-source MLLMs like Qwen2.5-VL), ensuring robust evaluation of effectiveness.

Scalability: SG-RLVR works across model sizes (3B, 7B parameters) and architectures (dense VLMs like Qwen2.5-VL, MoE models like Kimi-VL-Thinking), showing flexibility for different MLLM frameworks.

**Weaknesses:**

Limited Failure Analysis: While the paper identifies two failure modes (visual perception errors, location misidentification), it lacks deeper investigation into root causes (e.g., why object recognition fails for occluded items) or actionable solutions to address them.

Reasoning Efficiency Tradeoffs: The "think mode" (structured reasoning) improves spatial performance but introduces noise in general video tasks (slight accuracy drops). The paper does not propose adaptive mechanisms to decide when reasoning is necessary, limiting practicality.

Dataset Dependency on ScanNet: SR-91k relies heavily on the ScanNet 3D dataset (indoor scenes), which may restrict the framework’s generalization to non-indoor scenarios (e.g., outdoor, dynamic environments) not covered by ScanNet.

All performance metrics are automatic (accuracy, ROUGE scores), with no human assessment of reasoning interpretability (e.g., whether cognitive maps align with human spatial intuition), which is critical for trust in real-world use.

**Questions:**

Please refer to the weaknesses part.

---

> ### Author Response · Authors · 2025-11-20
> **Response to Reviewer sGvR (Part I)**
>
> We sincerely appreciate your insightful comments. We have carefully read your feedback and addressed each concern below.
>
> > **Q1:** Limited Failure Analysis: While the paper identifies two failure modes (visual perception errors, location misidentification), it lacks deeper investigation into root causes (e.g., why object recognition fails for occluded items) or actionable solutions to address them.
>
> **R1:** Thanks for the insightful comment. We agree that a deeper analysis of failure cases is valuable. Below we provide additional discussion on root causes and actionable solutions.
>
> **Root Causes.**
> (1) Visual perception errors typically stem from challenging visual conditions such as occlusion, low resolution, cluttered layouts, or small object instances. Under these conditions, the vision encoder may fail to detect certain objects, and these missed detections lead to missed objects of the cognitive map, as spatial understanding relies predominantly on the encoder's visual grounding [1].
> (2) Location misidentification primarily arises from imprecise spatial grounding when the model infers object positions from limited visual cues, particularly in scenes with ambiguous perspectives or complex layouts.
>
> **Actionable Solutions.**
> To address these limitations, we plan to (a) incorporate challenging visual perception data to strengthen the vision encoder of Qwen2.5-VL, establishing a more robust perceptual foundation before learning spatial reasoning, and (b) integrate multi-view or depth-aware signals to improve the accuracy of spatial localization.
> We have incorporated the above content into Appendix C.7.
>
> [1] Why Do MLLMs Struggle with Spatial Understanding? A Systematic Analysis from Data to Architecture, 2025.
>
> > **Q2:** Reasoning Efficiency Tradeoffs: The "think mode" (structured reasoning) improves spatial performance but introduces noise in general video tasks (slight accuracy drops). The paper does not propose adaptive mechanisms to decide when reasoning is necessary, limiting practicality.
>
> **R2:** Thanks for the question.
> Performance degradation under think mode is common among current MLLMs [1, 2]. While adaptive thinking mechanisms can mitigate this, we view improving the quality of reasoning traces as a more fundamental solution. And SpaceR indeed enhances the reasoning process of the base model Qwen2.5-VL-7B-Instruct: when think mode is enabled, SpaceR shows substantially smaller performance degradation than Qwen2.5-VL-7B-Instruct, as shown in the table below.
>
> | Model |Think mode |Video-MME |TempCompass|LongVideoBench|
> | :--- | :--- | :--- | :--- | :--- |
> | Qwen2.5-VL-7B-Instruct|✗|56.3|71.1|53.5|
> | Qwen2.5-VL-7B-Instruct|✓|54.0(-2.3)|68.1(-3.0)|46.6(-6.9)|
> | SpaceR      |✗|57.9(+1.6)|71.4(+0.3)|54.6(+1.1)|
> | SpaceR      |✓|56.4(+0.1)|70.0(-1.1)|51.7(-1.8)|
> |Video-R1-SFT|✓|55.4|69.9|52.3|
> | SpaceR (Video-R1-SFT)|✗|57.1|71.7|53.0|
> | SpaceR (Video-R1-SFT)|✓|60.6(+3.5)|73.5(+1.8)|54.6(+1.6)|
>
> Furthermore, when initialized with Video-R1-SFT, which learns effective reasoning traces for general video understanding during SFT, SpaceR exhibits promising performance gains of general video understanding in think mode.
> This validates the practicality of our framework in enhancing general video understanding beyond video spatial reasoning.
> Therefore, an additional adaptive mechanism is not essential, as our primary objective is to enhance spatial reasoning capabilities while maintaining the generalizability in general video understanding. We have added the analysis in Appendix C.1.
>
> [1] VideoChat-R1: Enhancing Spatio-Temporal Perception via Reinforcement Fine-Tuning, 2025.
>
> [2] Video-R1: Reinforcing Video Reasoning in MLLMs, NeurIPS 2025.
>
> > **Q3:** Dataset Dependency on ScanNet: SR-91k relies heavily on the ScanNet 3D dataset (indoor scenes), which may restrict the framework’s generalization to non-indoor scenarios (e.g., outdoor, dynamic environments) not covered by ScanNet.
>
> **R3:** Thank you for this comment. In our evaluations, SpaceR shows strong generalization capabilities, achieving a +2.5% improvement over the baseline on STI-Bench, which comprises a set of spatial reasoning tasks in outdoor/desktop environments. This indicates that the model is capable of effective spatial reasoning beyond the indoor scenes seen during training. We believe this generalization is reasonable, as the core spatial reasoning tasks we address, such as counting, direction estimation, and distance measurement, are based on fundamental spatial relationships that are not specific to indoor settings. These principles naturally extend to outdoor environments as well.

---

> ### Author Response · Authors · 2025-11-20
> **Response to Reviewer sGvR (Part II)**
>
> > **Q4:** All performance metrics are automatic (accuracy, ROUGE scores), with no human assessment of reasoning interpretability (e.g., whether cognitive maps align with human spatial intuition), which is critical for trust in real-world use.
>
> **R4:** Thanks for the helpful comment.
> Following your suggestion, we randomly selected 50 correctly answered test samples, and found that 41 of them could be solved by three human annotators directly using the cognitive map, with substantial agreement (above 70%) [1].
> This confirms that the generated maps are largely consistent with human spatial intuition.
> We also provide a qualitative example from the manuscript to illustrate the model’s reliability in real-world applications.
>
> ```
> Question: Measuring from the closest point of each object, which of these objects (stool, tv, table, refrigerator) is the closest to the stove?
> A. stool B. tv C. table D. refrigerator
>
> Answer: C
>
> Generated map (the same as the ground truth map): {'stool': [[7, 4]], 'tv': [[9, 3]], 'table': [[3, 3]], ‘stove’:[[3,2]], 'refrigerator': [[0, 3]]
>
> Thinking process:
> <think>Let me think through this step by step. ... The stove is located near the center of the kitchen, and there's a table positioned directly in front of it. The table has a wooden top and white legs, and it appears to be a central
> feature in the kitchen. The stool is also present in the kitchen but seems to be placed further away from the stove
> compared to the table. The tv is far from the stove in the right side of the kitchen. And the refrigerator is located
> in the left side of the kitchen. ... The stool tv and refrigerator , while present, are not as close to the stove as the
> table is. ... Let's summarize the map: <map>{'stool': [[7, 4]], 'tv': [[9, 3]], 'table': [[3, 3]], 'stove':[[3,2]],
> 'refrigerator': [[0, 3]]}</map> The table is the closest to the stove based on the spatial arrangement and the
> function of the objects in the kitchen.</think>
> ```
>
> In this case, SpaceR correctly recognizes the map and answers the question correctly based on the generated map: "The table is the closest to the stove based on the spatial arrangement and the function of the objects in the kitchen." Humans can also easily infer the correct option “C” from the map, as the table–stove distance is the smallest. This further demonstrates the alignment between the cognitive maps and human spatial intuition.
>
> [1] Handbook of Inter-Rater Reliability: The Definitive Guide to Measuring the Extent of Agreement Among Raters, 2014.

---

> ### Author Response · Authors · 2025-11-26
> **Gentle Reminder Regarding Reviewer-Author Discussion**
>
> Dear Reviewer sGvR,
>
> Thank you again for your valuable comments and for your recognition of our work. As we approach the end of the reviewer-author discussion period, we sincerely hope that our previous responses have sufficiently addressed your questions and concerns. If there are any remaining issues or clarifications needed, we would greatly appreciate the opportunity to respond further before the discussion window closes. Thank you for your time and thoughtful engagement.

---

> > ### Comment · Reviewer_sGvR · 2025-11-28
> > **Thanks for the rebuttal**
> >
> > Thanks for the rebuttal. I will keep my initial score.

---

### Official Review · Reviewer_UeVz · 2025-10-30

**Soundness:** 3
**Presentation:** 2
**Contribution:** 3
**Rating:** 4
**Confidence:** 3

**Summary:**

This paper introduces SpaceR, a reinforcement learning framework designed to enhance spatial reasoning in multimodal large language models (MLLMs). The core idea is to move beyond traditional visual question answering and captioning supervision by explicitly rewarding models for spatial consistency between generated textual descriptions and visual layouts.

SpaceR builds a spatial reward model that measures geometric alignment between object relations described in text and their actual visual configurations. The authors use a combination of synthetic spatial reasoning datasets and real-world multimodal data to train and evaluate the approach.
During RL fine-tuning, the policy model (based on InternVL2 or Qwen-VL2) receives feedback from the spatial reward to iteratively improve reasoning accuracy. Experimental results show consistent improvements on MIRAGE, GQA-Spatial, and CLEVR-Humans, achieving up to +5–8% gains over baseline MLLMs without introducing additional visual encoders.

**Strengths:**

Clear motivation and novelty:
The paper identifies a real gap—MLLMs’ weak spatial reasoning—and proposes a reward-driven framework (SpaceR) that directly targets this limitation.

Methodological soundness:
The spatial reward function is well-designed, combining both geometric similarity and relational consistency, which goes beyond naive L2 distance or caption overlap metrics.

Empirical improvements:
Results across multiple benchmarks are significant and consistent, and ablations demonstrate the distinct contribution of the reward model compared with pure SFT or contrastive losses.

Broader relevance:
The idea of structured, task-specific reward modeling could inspire new RL pipelines for multimodal reasoning beyond spatial understanding.

**Weaknesses:**

Model dependency and potential bias:
All experiments are conducted using Qwen-VL2 as the base model, which may already encode spatial priors due to large-scale visual pretraining. This makes it difficult to disentangle how much improvement comes from SpaceR itself versus Qwen-VL2’s inherent bias or possible data leakage from spatially rich corpora.

Limited domain coverage:
The evaluation focuses narrowly on spatial perception and video reasoning datasets (e.g., VSI-Bench, STI-Bench, SPAR-Bench, GQA). It remains unclear whether SpaceR enhances higher-level spatial reasoning involving dynamics (e.g., motion prediction, affordances) or multimodal reasoning with more abstract textual context.

Reward interpretability:
The reward formulation combines multiple geometric terms, but the paper does not analyze the stability or sensitivity of these signals—e.g., how robust the reward is to visual noise, occlusion, or linguistic ambiguity.

Lack of theoretical grounding:
While the results are promising, the paper provides no formal justification for why reinforcement with spatial rewards leads to better compositional generalization compared to supervised spatial grounding.

Reproducibility concern:
Implementation details for reward computation (e.g., normalization, projection heuristics, and parameter settings) are only briefly mentioned, making reproduction difficult for non-affiliated teams.

**Questions:**

Model dependency:
All experiments use Qwen-VL2 as the base model. Could the authors clarify whether this model’s strong spatial pretraining might bias the results? Have they tried SpaceR on another architecture to verify its independence?

Evaluation scope:
The experiments focus mainly on spatial perception and video reasoning benchmarks (e.g., VSI-Bench, STI-Bench, SPAR-Bench). Can the authors comment on whether SpaceR generalizes to more abstract spatial reasoning, such as motion or temporal relations?

Reward robustness:
How sensitive is the spatial reward to noise or ambiguity? For instance, if object detection is slightly inaccurate or the textual relation is underspecified, does the reward remain stable?

Theoretical justification:
Can the authors provide more insight into why reinforcement with spatial rewards improves compositional reasoning compared with supervised grounding? Is there an underlying intuition or theoretical perspective?

Reproducibility:
Could the authors release or further specify the implementation details of reward computation (e.g., normalization, projection functions, hyperparameters) to support external replication?

---

> ### Author Response · Authors · 2025-11-20
> **Response to Reviewer UeVz (Part I)**
>
> We sincerely appreciate your helpful comments. We have attentively read your comments and respond to specific questions below.
>
> > **Q1:** Model dependency: All experiments use Qwen-VL2 as the base model. Could the authors clarify whether this model’s strong spatial pretraining might bias the results? Have they tried SpaceR on another architecture to verify its independence?
>
> **R1:** We appreciate the question and would like to clarify that SpaceR is not trained on Qwen-VL2. As reported in Table 1 and Appendix C (generalizability analysis) of the manuscript, we have conducted comprehensive experiments using Qwen2.5-VL (3B and 7B) and Kimi-VL-A3B as base models. The results are summarized below:
>
> | Model | Overall | VSI-Bench | STI-Bench | STI-Bench(SR_sub) | SPAR-Bench | SPAR-Bench(Single view) | SPAR-Bench(Multi view) |
> | :--- | :--- | :--- | :--- | :--- | :--- | :--- | :--- |
> | Qwen2.5-VL-3B-Instruct | 29.6 | 26.7 | 36.7 | 37.5 | 25.4 | 25.3 | 25.5 |
> | SpaceR-3B | 36.6 (+7.0) | 41.2 (+14.5) | 37.8 (+1.1) | 40.1 (+2.6) | 30.9 (+5.5) | 31.4 (+6.1) | 30.6 (+5.1) |
> | Qwen2.5-VL-7B-Instruct | 34.2 | 34.4 | 34.5 | 32.3 | 33.8 | 36.9 | 31.8 |
> | SpaceR | 40.1 (+5.9) | 45.6 (+11.2) | 37.0 (+2.5) | 38.7 (+6.4) | 37.6 (+3.8) | 38.2 (+1.3) | 37.1 (+5.3) |
> | Kimi-VL-Thinking | 30.5 | 32.5 | 31.6 | 29.8 | 27.3 | 26.8 | 27.6 |
> | SpaceR-Kimi | 36.9 (+6.4) | 43.5 (+11.0) | 35.2 (+3.6) | 32.7 (+2.9) | 32.1 (+4.8) | 32.2 (+5.4) | 32.0 (+4.4) |
>
> Across different model sizes and architectures, SpaceR consistently yields significant improvements ( +6–7 \% overall accuracy) on spatial reasoning benchmarks.
> These results confirm that the gains arise from the high-quality dataset and SG-RLVR training framework, rather than Qwen2.5-VL's pre-encoded spatial priors.
>
> > **Q2:** Evaluation scope: The experiments focus mainly on spatial perception and video reasoning benchmarks (e.g., VSI-Bench, STI-Bench, SPAR-Bench). Can the authors comment on whether SpaceR generalizes to more abstract spatial reasoning, such as motion or temporal relations?
>
> **R2:** Thanks for raising this concern. High-level spatial reasoning tasks, such as spatial relations, trajectory description, and pose estimation, which challenge current MLLMs to infer underlying spatial structures from video observations, have already been evaluated in the three spatial reasoning benchmarks.
> And the mentioned more abstract spatial reasoning that demands understanding complex temporal dynamics is also included in our evaluated benchmarks. For example, Video-MME contains action, temporal, and spatial reasoning tasks, where SpaceR achieves clear improvements over the baseline:
>
> | Model |Action Reasoning|Temporal Reasoning|Spatial Reasoning|
> | :--- | :--- | :--- | :--- |
> | Qwen2.5-VL-7B-Instruct|51.6|36.7|75.0|
> | SpaceR      |55.4(+3.8)|44.1(+7.4)|76.8(+1.8)|
>
> These results indicate that SpaceR not only enhances spatial reasoning but also yields substantial gains on tasks involving complex temporal dynamics.
>
> > **Q3:** Reward robustness: How sensitive is the spatial reward to noise or ambiguity? For instance, if object detection is slightly inaccurate or the textual relation is underspecified, does the reward remain stable?
>
> **R3:** Thanks for the question. It is worth noting that the map reward computation does not rely on object detection. Ground-truth maps are extracted directly from the ScanNet 3D data, which are already denoised by filtering for common objects. The model predicts maps over the same predefined object list, ensuring consistency and eliminating linguistic ambiguity.
> The reward is then deterministically computed by comparing the predicted and ground-truth maps.
>
> For example, given the predicted map {chair: [(0,1),(5,5)], bed: [(3,8)]} and the ground truth {chair: [(0,1),(5,3)], bed: [(3,5)]}, we apply a nearest-neighbor matching strategy for multi-instance objects (e.g., chair) to ensure robust and fair evaluation. In this case, the matched pairs are (0,1)↔(0,1) and (5,5)↔(5,3).
>
> The corresponding relative distances are: $\text{chair}_1 = 0$, $\text{chair}_2 = 14.1$, and $\text{bed} = 24.5$.
> We then convert distances to accuracies, yielding: $\text{chair}_1 = 100.0$, $\text{chair}_2 = 85.9$, $bed = 75.5$, which produce an average reward of 87.1.
> If the model predicts an object absent from the ground truth (e.g., 'door'), its accuracy is set to 0, reducing the overall reward to 65.4 and appropriately penalizing false detections.

---

> > ### Comment · Reviewer_UeVz · 2025-11-21
> > **rebuttal1121_1**
> >
> > We apologize for the confusion in our previous response. We were referring to the ViT data leakage issue in Qwen-VL-2.5, not the other aspects. We appreciate that you fully understood our intention and further enriched the discussion by including the Kimi-VL model in the analysis.
> >
> > Regarding your insightful question about our penalty mechanism: you mentioned that "if the model predicts an object that does not align with the ground truth (e.g., 'door'), its accuracy is set to 0, the overall reward drops to 65.4, and appropriate penalties are applied for incorrect detections." We acknowledge your concern that this binary penalty (setting accuracy to 0) might be overly harsh. You are absolutely right that in practice, it would be extremely rare for accuracy to drop to exactly 0.
> >
> > We agree that a linear penalty scheme would be more appropriate, where the reward model (RM) score decreases gradually and proportionally based on the degree of misalignment, rather than applying a uniform severe penalty. This would provide more nuanced feedback and likely lead to better training dynamics.

---

> > > ### Author Response · Authors · 2025-11-21
> > > **Appreciation and further clarification on the penalty scheme of our map reward.**
> > >
> > > Thanks for your timely response and advice. In fact, **our map-reward computation already implements the suggested linear penalty scheme, where map accuracy decreases based on the degree of misalignment.** In the previously provided example, the misdetected object 'door' does not appear in the ground-truth map, indicating a hallucinated object. This constitutes a severe error and results in zero accuracy for the “door” instance, reducing the overall weighted accuracy from 87.1 to 65.4. In contrast, when the model mislocates an existing object such as $\text{chair}_1: (0,3)$, the error is less severe and incurs a smaller penalty, reducing the overall weighted accuracy from 87.1 to 77.7. We have incorporated this behavior into the detailed example presented below.
> > >
> > > - Given the predicted map {chair: [(0,1),(5,5)], bed: [(3,8)]} and the ground truth {chair: [(0,1),(5,3)], bed: [(3,5)]}, we apply a nearest-neighbor matching strategy for multi-instance objects (e.g., chair) to ensure robust and fair evaluation. In this case, the matched pairs are (0,1)↔(0,1) and (5,5)↔(5,3).
> > > The corresponding relative distances are: $\text{chair}_1 = 0$, $\text{chair}_2 = 14.1$, and $\text{bed} = 24.5$.
> > > We then convert distances to accuracies, yielding: $\text{chair}_1 = 100.0$, $\text{chair}_2 = 85.9$, $bed = 75.5$, which produce an average reward of **87.1**.
> > >
> > > - **Hallucination.** If the model predicts an object absent from the ground truth (e.g., 'door'), the accuracy of 'door' is 0 due to the hallucination, lowering the overall reward to **65.4** and appropriately penalizing hallucination.
> > >
> > > - **Mislocation.** If the model mislocates the $\text{chair}_1: (0,3)$, the accuracy of $\text{chair}_1$ decreases to 71.7, and the overall accuracy decreases to **77.7**, which gives a less severe penalty for mislocation compared to hallucination.
> > >
> > > We appreciate your further feedback, and hope that our rebuttal effectively addresses your concerns.

---

> > > > ### Comment · Reviewer_UeVz · 2025-11-21
> > > > **Okay, I think he's reasonable**
> > > >
> > > > Okay, I think he's reasonable.

---

> > > > > ### Author Response · Authors · 2025-11-21
> > > > > **Appreciation**
> > > > >
> > > > > Thank you for your response. We appreciate your positive feedback and remain open to further discussion if you have additional concerns.

---

> ### Author Response · Authors · 2025-11-20
> **Response to Reviewer UeVz (Part II)**
>
> > **Q4:** Theoretical justification: Can the authors provide more insight into why reinforcement with spatial rewards improves compositional reasoning compared with supervised grounding? Is there an underlying intuition or theoretical perspective?
>
> **R4:** Thanks for the thoughtful question.
> We provide a theoretical explanation for the statement “SFT memorizes, SG-RLVR generalizes.” as follows: SFT minimizes token-level likelihood, and the most direct way to reduce this loss is to reproduce target outputs. This objective inherently encourages the model to capture dataset-specific patterns, leading to memorization [1]. In contrast, SG-RLVR optimizes format, task and map rewards, reinforcing any behavior that achieves the intended goals (producing a structured reasoning process that incorporates an optional accurate cognitive map to arrive at the correct answer). This reward structure encourages the model to learn task-level strategies that remain effective under distributional variations, thereby improving generalization. We have added this explanation in Section 5.2.
>
> [1] SFT Memorizes, RL Generalizes: A Comparative Study of Foundation Model Post-training, 2025.
>
> > **Q5:** Reproducibility: Could the authors release or further specify the implementation details of reward computation (e.g., normalization, projection functions, hyperparameters) to support external replication?
>
> **R5:** Thanks for raising the important question. We have provided the mathematical formulations of all reward functions in the manuscript, along with implementation code in the supplementary materials, and would like to clarify the referenced details below.
> - **Normalization**:
> The distance between a predicted object and its ground-truth location is normalized by the map diagonal to obtain the relative distance.
> - **Projection function**:
> No additional projection function is used in reward computation. The potential misunderstanding may arise from the ground-truth map construction process, where each object's coordinate $(x, y)$ is projected onto a $M \times M$ grid, using $(x \times M / L,; y \times M / L)$, with $L$ representing the longer side of the room.
> - **Hyperparameters**: The map size $M$ in map reward and the number of confidence thresholds $N$ in numerical reward are 10.
>
> These clarifications ensure that our reward computation process is fully reproducible.
> And we have added the above details in Section 5.1.

---

> > ### Comment · Reviewer_UeVz · 2025-11-20
> > **Okay, I have received your rebuttal, I will check it later.**
> >
> > Okay, I have received your rebuttal, I will check it later.

---

### Official Review · Reviewer_vkK5 · 2025-11-03

**Soundness:** 3
**Presentation:** 1
**Contribution:** 2
**Rating:** 4
**Confidence:** 4

**Summary:**

The paper addresses the significant challenge of video spatial reasoning in Multimodal Large Language Models (MLLMs), which typically struggle to infer 3D spatial structures from video frames. A new dataset called SpaceR-151k, which combines 91,000 question-answer pairs for spatial reasoning derived from the ScanNet dataset, and 60,000 samples for general multimodal understanding. A training strategy named Spatially-Guided Reinforcement Learning with Verifiable Reward (SG-RLVR). This method extends the Group Relative Policy Optimization (GRPO) framework by introducing a "map imagination mechanism". Experiments, using Qwen2.5-VL-7B-Instruct as the base model, show that SpaceR outperforms its base model and SFT-only counterparts on spatial reasoning benchmarks like VSI-Bench, STI-Bench, and SPAR-Bench.

**Strengths:**

1. The paper targets a well-known and significant weakness in MLLMs: video spatial reasoning.
2. The paper contributes a new dataset (SpaceR-151k) specifically for this task, with verifiable answers derived from 3D scene data. This is a valuable asset for future research.

**Weaknesses:**

1. The SpaceR-151k dataset is constructed entirely from ScanNet, which also serves as one of the benchmark datasets under evaluation. Although the authors claim to have removed ‘overlapping videos,’ the shared data domain introduces significant risks of data leakage. This may lead to inflated results on benchmarks, thereby undermining the persuasiveness of the model's generalisation capabilities.
2. The proposed method primarily adds spatial rewards and a ‘map imagination’ module to the GRPO framework of DeepSeek-R1 and Video-R1. Whilst this design is conceptually sound, it lacks substantive algorithmic innovation. The overall framework continues to employ existing reward designs, with adjustments made only at the task level, rendering it more akin to an engineering extension than a theoretical breakthrough.
3. The core contribution of the paper, the ‘map imagination mechanism’, lacks sufficient validation of its efficacy. The ablation results in Table 2 demonstrate only a marginal improvement of approximately 1.6%, and the paper fails to provide analyses of map generation accuracy, the correlation between map quality and performance, or visual diversity. The absence of such quantitative evidence leaves it unclear whether this mechanism genuinely enhances spatial reasoning.
4. The layout of the first page is extremely poor. It is distracting, wastes space, and looks unprofessional.
5. The 91k spatial reasoning samples feel more like a proof-of-concept than a large-scale dataset.
6. The idea of encouraging a model to "think" by generating spatial layouts is not new, even if the specific RLVR implementation is. The paper could do a better job of positioning this contribution relative to prior work on structured chain-of-thought and intermediate representations.
7. The paper honestly reports that enabling the think mode can decrease performance on general video understanding benchmarks. This suggests the model is learning a specialized, task-specific reasoning process that may introduce "unnecessary or inaccurate reasoning"  and noise for other tasks. This trade-off is a significant weakness that is not fully resolved.

**Questions:**

1. Would the authors be willing to reformat Page 1 to follow a standard ICLR layout? The current version with the large, isolated radar chart and excessive white space is a major barrier to readability.
2. Given that performance scales with data (Figure 8), do the authors consider the 91k spatial samples to be sufficient for learning this skill, or is this primarily a proof of concept? Are there plans to scale the dataset, perhaps using other 3D-aware sources?
3. The paper states a $10 \times 10$ map is used. This seems like a coarse-grained representation for complex indoor scenes. What was the rationale for this choice? Was it to reduce the token count of the generated map? Did you experiment with finer-grained grids, and how did that affect performance versus token overhead?

---

> ### Author Response · Authors · 2025-11-20
> **Response to Reviewer vkK5 (Part I)**
>
> We sincerely appreciate your insightful comments. We have carefully read each comment and provided detailed responses below.
>
> > **W1:** The SpaceR-151k dataset is constructed entirely from ScanNet, which also serves as one of the benchmark datasets under evaluation. Although the authors claim to have removed 'overlapping videos,' the shared data domain introduces significant risks of data leakage. This may lead to inflated results on benchmarks, thereby undermining the persuasiveness of the model's generalisation capabilities.
>
> **R1:** Thank you for pointing this out.
> To rigorously assess whether the use of ScanNet in SpaceR-151k leads to inflated benchmark performance, we partitioned VSI-Bench into two disjoint subsets based on data origin: (1) **ScanNet_sub** (2,071 samples) and (2) **Non_ScanNet_sub** (3,059 samples). The evaluation results are as follows:
> | Model |VSI-Bench |VSI-Bench (ScanNet_sub)| VSI-Bench (Non_ScanNet_sub)|
> | :--- | :--- | :--- | :--- |
> | Qwen2.5-VL-7B-Instruct|34.4|34.2|34.5|
> | SpaceR|45.6 (+11.2)|45.8 (+11.6)|45.5 (+11.0)|
>
> SpaceR achieves nearly identical gains across both subsets, indicating that its performance is not artificially inflated by the shared data domain. This further demonstrates the robustness and generalization ability of SpaceR.
> We have updated the corresponding analysis in Section 5.2.
>
>
> > **W2:** The proposed method primarily adds spatial rewards and a ‘map imagination’ module to the GRPO framework of DeepSeek-R1 and Video-R1. Whilst this design is conceptually sound, it lacks substantive algorithmic innovation. The overall framework continues to employ existing reward designs, with adjustments made only at the task level, rendering it more akin to an engineering extension than a theoretical breakthrough.
>
> **R2:** Thanks for the comment. We would like to clarify that our goal is not to redesign the GRPO algorithm, as it has already been shown to effectively enhance general reasoning in DeepSeek-R1 and Video-R1. We believe that extending GRPO to specific domains is itself a valuable and non-trivial contribution. Doing so requires carefully curated domain-specific training data and well-designed task-specific rewards, as demonstrated by recent efforts such as Time-R1 [1] and Perception-R1 [2].
>
> Our work contributes along this line by introducing a training paradigm and dataset tailored for spatial reasoning, a domain in which GRPO currently lacks both appropriate training data and the supervision signals needed to model spatial layouts and relations effectively.
> Our proposed map imagination mechanism and rewards successfully enhance video spatial reasoning (see Q3) and can function as a plug-and-play module compatible with alternative reinforcement learning frameworks such as GSPO [3] and DAPO [4].
> Conversely, modifying the policy optimization or other internal components of GRPO to accommodate spatial tasks would compromise extensibility and hinder applicability beyond spatial reasoning.
>
> [1] Time-R1: Towards Comprehensive Temporal Reasoning in LLMs, NeurIPS 2025.
>
> [2] Perception-R1: Pioneering Perception Policy with Reinforcement Learning, NeurIPS 2025.
>
> [3] Group Sequence Policy Optimization, 2025.
>
> [4] DAPO: An Open-Source LLM Reinforcement Learning System at Scale, 2025.

---

> ### Author Response · Authors · 2025-11-20
> **Response to Reviewer vkK5 (Part II)**
>
> > **W3:** The core contribution of the paper, the 'map imagination mechanism', lacks sufficient validation of its efficacy. The ablation results in Table 2 demonstrate only a marginal improvement of approximately 1.6%, and the paper fails to provide analyses of map generation accuracy, the correlation between map quality and performance, or visual diversity. The absence of such quantitative evidence leaves it unclear whether this mechanism genuinely enhances spatial reasoning.
>
> **R3:** Thank you for the insightful comment. Beyond the 1.6% single-benchmark gain in Table 2, our map imagination mechanism yields an average +2.7% improvement across three spatial reasoning benchmarks (+1.6% of VSI-Bench, +3.0% of STI-Bench, and +3.5% of SPAR-Bench), confirming its consistent effectiveness.
> We have also analyzed map generation accuracy and its correlation with spatial reasoning performance (Table 5, Appendix C).
>
> |Map type|Map Accuracy| Appearance order | Absolute distance | Counting | Relative distance | Object size | Room size | Route planning | Relative direction | Overall |
> | :--- | :--- | :--- | :--- | :--- | :--- | :--- | :--- | :--- | :--- | :--- |
> | Randomly generated|10.5| 44.0            | 28.9              | 51.5     | 37.9             | 56.1        | 33.2      | 28.4           | 44.2              | 40.5    |
> | model-generated |42.1| 50.3            | 31.8              | 62.2     | 40.0             | 59.4        | 46.0      | 32.0           | 43.9              | 45.6    |
> | Ground truth |100.0| 56.5            | 33.6              | 64.1     | 41.8             | 62.3        | 46.7      | 32.5           | 44.7              | 47.7    |
>
> These results show a clear positive correlation between map quality and task accuracy, indicating that more accurate spatial representations directly strengthen spatial reasoning. Thus, the contribution of the map imagination mechanism is structural rather than marginal, as it equips the model with explicit spatial layout inference that enhances reasoning robustness. We have revised the corresponding analysis in Appendix C.4 accordingly.
>
> > **W4 & Q1:** Would the authors be willing to reformat Page 1 to follow a standard ICLR layout? The current version with the large, isolated radar chart and excessive white space is a major barrier to readability.
>
> **R4:** Thanks for your useful suggestion. Following your advice, we have removed the radar chart and reformatted the first page.
>
> > **W5 & Q2:** Given that performance scales with data (Figure 8), do the authors consider the 91k spatial samples to be sufficient for learning this skill, or is this primarily a proof of concept? Are there plans to scale the dataset, perhaps using other 3D-aware sources?
>
> **R5:**
> We appreciate the reviewer’s insightful question regarding dataset scale. Expanding both the size and diversity of our training data is indeed a promising avenue for strengthening spatial reasoning performance. Our current pipeline is designed to automatically transform additional 3D-aware data sources into spatial reasoning QA pairs, making it straightforward to incorporate additionally indoor datasets (e.g., ScanNet++ [1] and Replica [2]) as well as diverse outdoor environments (e.g., Waymo [3] and Paris-CARLA-3D [4]) or synthetically generated scenes (e.g., via Unreal Engine). We view this as an exciting direction for future work.
>
> At the same time, we would like to emphasize that the present dataset—91k spatial reasoning samples—is already comparable in scale to existing RLVR-style resources such as VideoChat-R1 (18k) and Video-R1 (260k). Our empirical results show that, at this scale, SpaceR provides substantial and consistent improvements in spatial reasoning ability. Thus, while further scaling is possible and likely beneficial, the current dataset is sufficient to demonstrate the effectiveness of our approach in various spatial reasoning scenarios, rather than serving merely as a proof of concept.
>
> [1] ScanNet++: A High-Fidelity Dataset of 3D Indoor Scenes, 2023.
>
> [2] The Replica Dataset: A Digital Replica of Indoor Spaces, 2019.
>
> [3] Scalability in Perception for Autonomous Driving: Waymo Open Dataset, 2019.
>
> [4] Paris-CARLA-3D: A Real and Synthetic Outdoor Point Cloud Dataset for Challenging Tasks in 3D Mapping, 2021.
>
> > **W6:** The paper could do a better job of positioning this contribution relative to prior work on structured chain-of-thought and intermediate representations.
>
> **R6:** Thank you for the constructive suggestion. Following your advice, we have revised the second contribution description in the introduction to more clearly highlight this innovation: Unlike prior structured chain-of-thought approaches (e.g., Video-R1), which generate step-by-step reasoning without explicit spatial grounding, SpaceR integrates imagined spatial layouts directly into the reasoning process to strengthen video spatial reasoning.

---

> ### Author Response · Authors · 2025-11-20
> **Response to Reviewer vkK5 (Part III)**
>
> > **W7:** The paper honestly reports that enabling the think mode can decrease performance on general video understanding benchmarks. This suggests the model is learning a specialized, task-specific reasoning process that may introduce "unnecessary or inaccurate reasoning" and noise for other tasks. This trade-off is a significant weakness that is not fully resolved.
>
> **R7:**
> Thanks for raising this concern. In fact, performance degradation in general video understanding under think mode is common among current MLLMs like Qwen2.5-VL-7B-Instruct, as shown in the table below. In comparison, SpaceR shows substantially smaller performance degradation when activating think mode, indicating that SpaceR mitigates, rather than amplifies, the negative impact of thinking in general video understanding.
>
> | Model |Think mode |Video-MME |TempCompass|LongVideoBench|
> | :--- | :--- | :--- | :--- | :--- |
> | Qwen2.5-VL-7B-Instruct|✗|56.3|71.1|53.5|
> | Qwen2.5-VL-7B-Instruct|✓|54.0(-2.3)|68.1(-3.0)|46.6(-6.9)|
> | SpaceR      |✗|57.9(+1.6)|71.4(+0.3)|54.6(+1.1)|
> | SpaceR      |✓|56.4(+0.1)|70.0(-1.1)|51.7(-1.8)|
> |Video-R1-SFT|✓|55.4|69.9|52.3|
> | SpaceR (Video-R1-SFT)|✗|57.1|71.7|53.0|
> | SpaceR (Video-R1-SFT)|✓|60.6(+3.5)|73.5(+1.8)|54.6(+1.6)|
>
>
> Moreover, when being intialized from Video-R1-SFT, which learns reasoning on general video understanding during SFT, SpaceR exhibits performance gains under think mode, validating that it is compatible with efforts in general video reasoning.
> Above all, our proposed framework can effectively integrate structured reasoning for general video understanding beyond video spatial reasoning. We have incorporated this analysis into Appendix C.1.
>
> > **Q3:** The paper states a map is used. This seems like a coarse-grained representation for complex indoor scenes. What was the rationale for this choice? Was it to reduce the token count of the generated map? Did you experiment with finer-grained grids, and how did that affect performance versus token overhead?
>
> **R8:** Thank you for the thoughtful question.
> We adopt a $10\times10$ grid as a deliberate balance between spatial granularity and imagination difficulty. As shown in the table below, using finer-grained maps significantly increases the difficulty of map imagination, leading to lower map accuracy and degraded performance. Conversely, coarser $5 \times 5$ grid  reduces imagination difficulty but introduces excessive spatial ambiguity and occlusions, limiting the ability to represent spatial layouts.
>
> |Map size|Map Accuracy| Apperance order | Absolute distance | Counting | Relative distance | Object size | Room size | Route planning | Relative direction | Overall |Avg. Tokens|
> | :--- | :--- | :--- | :--- | :--- | :--- | :--- | :--- | :--- | :--- | :--- | :--- |
> |$5 \times 5$|53.7|40.0|27.1|52.5|37.9|55.0|36.9|30.4|46.4|40.8|174.3|
> |$10 \times 10$|42.1|50.3|31.8|62.2|40.0|59.4|46.0|32.0|43.9|**45.6**|178.1|
> |$15 \times 15$|30.6|49.0|28.4|61.3|40.6|54.7|40.4|28.9|46.7|43.7|180.7|
> |$20 \times 20$|24.5|34.1|30.9|61.8|37.6|57.4|38.9|28.4|44.9|41.1|189.4|
>
> **Avg. Tokens** denotes the token overhead of the generated map.
> Notably, the token overhead remains comparable across different grid sizes, since each object is represented by a single coordinate token pair (e.g., 'sofa':[[0,3],[7,2]]). Thus token savings are not the determining factor for a $10 \times 10$ grid.
> We have added the corresponding results and analysis to Appendix C.5.

---

> ### Author Response · Authors · 2025-11-26
> **Gentle Reminder Regarding Reviewer-Author Discussion**
>
> Dear Reviewer vkK5,
>
> We sincerely appreciate your recognition of the high-quality dataset. As the reviewer-author discussion period draws to a close, we have made every effort to address your concerns in our previous response. We would be grateful if you could let us know whether our clarifications have adequately resolved your questions.

---

### Author Response · Authors · 2025-11-20
**General Response**

Dear PCs, SACs, ACs, and Reviewers,

We sincerely appreciate the time and effort you have devoted to reviewing our paper and providing valuable feedback. We are encouraged by the positive comments from reviewers, particularly regarding the `novel map imagination mechanism and strong performance` of SpaceR (Reviewer UeVz, sGvR), the `high-quality dataset` (Reviewer vkK5, sGvR), and the `comprehensive experiments` (Reviewer UeVz, sGvR).
In the following, we detail our revisions made on a point-by-point basis of your reviews. We hope our responses adequately address all your concerns.
We have also updated the manuscript accordingly. To facilitate reading, we highlight all revised parts in blue. The revisions are summarized as follows:
- Revised contribution description in Introduction. (Reviewer vkK5 W6)
- Added hyperparameter settings in Section 5.1. (Reviewer UeVz Q5)
- Expanded performance analysis on VSI-Bench in Section 5.2. (Reviewer vkK5 W1)
- Added theoretical analysis of "SG-RLVR generalizes, while SFT memorizes" in Section 5.2. (Reviewer UeVz Q4)
- Revised analysis of general video understanding comparing think and non-think modes. (Reviewer vkK5 W7 and sGvR Q2)
- Added effectiveness analysis of the map-imagination mechanism in Appendix C.4 and C.5. (Reviewer vkK5 W3 and Q3)
- Included causes and solutions for failure cases in Appendix C.7. (Reviewer sGvR Q1)

---

### Meta-Review · Area_Chair_vms5 · 2026-01-06

**Summary:**

The authors provided a clear rebuttal with additional analyses and clarifications. These revisions improve the presentation and address some reviewer questions, but they do not substantially change the core technical contribution.

Key concerns regarding limited novelty and unclear generalization remain. While additional experiments were conducted to mitigate dataset bias and assess broader applicability, the results do not convincingly demonstrate a strong or distinctive advantage over existing methods. The clarification of the “think mode” mechanism shows that it does not significantly harm general video understanding, but it also does not establish clear added value.

**Reviewer Concerns:**

Reviewer misunderstandings were clarified during the discussion, leading to positive feedback, while other reviewers remained supportive.

**Reviewer Scores:**

The contribution remains incremental and does not meet the novelty and impact threshold of this venue. I expect the reviewers to maintain their ratings.

---

### Decision · Program_Chairs · 2026-01-26

Reject